# MedMamba: Multi-View State Space Models with Adaptive Graph Learning for Medical Time Series Classification

Da Zhang [1 2 *]  Bingyu Li [2]  Zhiyuan Zhao [2]  Hongyuan Zhang [3]  Junyu Gao [2 1 †]  Xuelong Li [2 †]

## Abstract

Medical time series are central to healthcare, enabling continuous monitoring and supporting timely clinical decisions. Despite recent progress, existing methods struggle to jointly model local-global dynamics and handle nonstationarities like baseline drift, while often failing to capture latent channel interactions. To address these challenges, we propose **MedMamba**, an end-to-end architecture that integrates state space models with domain-specific inductive biases. Specifically, MedMamba first *employs multi-scale convolutional embeddings* to capture discriminative local morphology. Second, to mitigate nonstationarity, we introduce a *tri-branch differential state space encoder* that processes raw, temporal-difference, and frequency-domain views, fusing them to emphasize informative patterns while suppressing drift. Furthermore, to uncover latent channel correlations, we design a *spatial graph Mamba module* that learns a directed dependency structure regularized toward sparsity and acyclicity, which obviates the need for predefined graphs. Extensive experiments on five real-world datasets demonstrate that MedMamba achieves state-of-the-art performance while maintaining linear computational complexity, and ablation studies validate each component's contribution. Code is available at https://github.com/zhangda1018/MedMamba.

## 1. Introduction

Medical time series (MedTS) are critical in modern healthcare (Yuan et al., 2025; Chen et al., 2024c). Continuous physiological recordings support early detection of abnor-

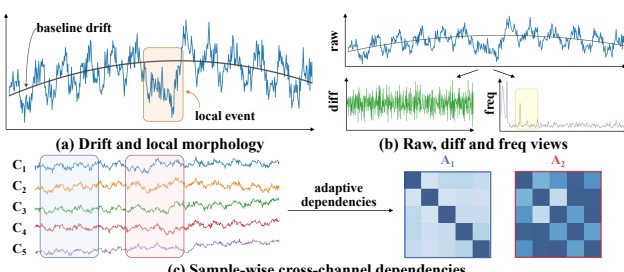

*Figure 1.* Key challenges in MedTS classification. (a) Baseline drift can obscure discriminative local morphology. (b) Complementary raw, temporal difference, and frequency-domain views provide robust cues under drift. (c) Cross-channel dependencies vary across segments. Dependency matrices ($A_1$, $A_2$) are from two highlighted segments extracted from the same recording.

malities, longitudinal monitoring, and decision support. Examples include electroencephalography (EEG) and electrocardiography (ECG) signals (Tang et al., 2025; Chen et al., 2025), where patterns in waveforms and rhythms can reveal neurological and cardiovascular conditions. Accurate classification can improve triage, diagnosis, and personalized treatment planning (Liu et al., 2025a; Deng et al., 2025).

Because of domain-specific characteristics, MedTS classification remains challenging (Yisimitila et al., 2025), as illustrated in Figure 1. Clinically relevant information may appear as transient events, persistent trends (Luo et al., 2024), or long-duration dependencies, requiring models to integrate local morphology with long context (Chen et al., 2024b). Most recordings are multichannel, and diagnostic cues can depend on segment-wise cross-channel interactions (Chen et al., 2024a). Moreover, real recordings are rarely stationary (Wen et al., 2023). Baseline drift and acquisition artifacts introduce slow shifts that can mask discriminative structure and reduce generalization, especially under subject-independent evaluation (Zhang et al., 2025).

Prior work has attempted to address these challenges but often faces trade-offs (Liu et al., 2024c). Deep learning methods utilize convolutional architectures to capture local morphology (Lawhern et al., 2018; Miltiadous et al., 2023a) or attention mechanisms for long-range modeling (Wang et al., 2024). Graph-based methods explicitly model inter-channel dependencies through learned structures (Luo et al.,

*Completed during internship at TeleAI. [1]Northwest Polytechnical University [2]TeleAI, China Telecom [3]University of Hong Kong. Correspondence to: Junyu Gao <gjy3035@gmail.com>, Xuelong Li <xuelong_li@ieee.org>.

*Proceedings of the 43rd International Conference on Machine Learning*, Seoul, South Korea. PMLR 306, 2026. Copyright 2026 by the author(s).

2025; Fan et al., 2025). While these families have advanced performance, they struggle to balance expressive temporal modeling with adaptive multichannel interactions. Moreover, maintaining robustness to drift without compromising efficiency on long recordings remains challenging. (Liu et al., 2025b).

State space models like Mamba (Gu & Dao, 2024) provide an appealing alternative backbone, which models long context with linear complexity and avoids the quadratic cost of Transformers (Smith et al., 2023). While recent adaptations like EEGMamba (Gui et al., 2024) and TSC-Mamba (Ahamed & Cheng, 2025) have explored biosignal applications, standard SSMs primarily focus on global sequence modeling. They lack inherent inductive biases to capture scale-variant local morphology, remain sensitive to low-frequency nonstationarities like baseline drift, and process channels independently without explicitly modeling the dynamic spatial topology required for multichannel MedTS.

In this paper, we propose MedMamba[1], an end-to-end architecture that integrates state space models with inductive biases tailored to these challenges. Specifically, to bridge the gap between local detail and global context, MedMamba employs multi-scale convolutional embeddings (MCE) that capture discriminative local morphology. To mitigate nonstationarity, we introduce a tri-branch differential state space encoder (TDSSE) that processes raw, temporal-difference, and frequency-domain views, fusing them to emphasize informative patterns while suppressing drift. Furthermore, to uncover latent sensor correlations without relying on predefined graphs, we design a spatial graph Mamba module (SGM). This module learns a directed dependency structure regularized toward sparsity and acyclicity, which enables the model to capture complex cross-channel interactions purely from data.

We evaluate MedMamba on five real-world multichannel datasets under both sample-based and subject-based protocols. The results show consistent improvements over strong baselines with linear complexity in sequence length, and ablations confirm the contribution of each component. Our main contributions can be summarized as follows:

- **MedMamba architecture**. We propose MedMamba, an architecture that combines state space models with domain-specific inductive biases to enable efficient and robust MedTS classification.

- **Tri-branch differential state space encoder**. We introduce a tri-branch differential state space encoder that leverages raw, differential, and frequency-domain views to mitigate non-stationarity and improve robustness to baseline drift.

- **Spatial graph mamba module**. We design a graph Mamba module that dynamically learns latent, adaptive spatial dependencies among sensors, capturing complex cross-channel interactions without requiring any prior topological knowledge.

- **Extensive real-world evaluation**. Extensive experiments on five real-world medical datasets demonstrate that MedMamba achieves SOTA performance while maintaining linear computational complexity.

## 2. Related Work

### 2.1. Deep Learning for Medical Time Series

Deep learning is the dominant paradigm for MedTS classification (Sun et al., 2020). Early CNN-based methods extract local morphological features (Tan et al., 2020; Wang et al., 2023); for example, EEGNet (Lawhern et al., 2018) introduced compact depthwise separable convolutions to efficiently learn temporal and spatial filters. To capture long-range dependencies beyond CNNs, Transformer-based architectures such as Medformer (Wang et al., 2024) and other time-series Transformers (Nie et al., 2023; Liu et al., 2024b) use self-attention to model global context. However, Transformers scale quadratically with sequence length, limiting applicability to high-resolution physiological recordings (Zhou et al., 2021; Yuan et al., 2025). Distinct from these approaches, MedMamba leverages a state space backbone for linear complexity while modeling global context, and our tri-branch encoder integrates differential and frequency-domain views to combat non-stationarity and baseline drift.

### 2.2. State Space Models for Time Series

State Space Models (Gu et al., 2022; Smith et al., 2023) have gained attention for modeling extremely long sequences with linear scaling. Mamba (Gu & Dao, 2024) further introduces selective scanning to compress context into a fixed-size state while dynamically filtering inputs. While SSMs have been explored for general time series forecasting (Wang et al., 2025) and classification (Schiff et al., 2024), their application to MedTS is emerging (Gui et al., 2024). However, standard Mamba lacks dedicated mechanisms for key MedTS challenges such as severe non-stationarity (Liu et al., 2022; Ryan et al., 2025) and multi-channel interactions (Ekambaram et al., 2023). Different from these lines, MedMamba targets robust classification by explicitly integrating multi-views and coupling them with sample-adaptive graph learning for cross-channel dependencies.

### 2.3. Graph Learning of Spatial Dependencies

MedTS are multivariate, and interactions among sensors carry critical diagnostic information (Li et al., 2023). Graph

---

[1] We note that this name has been previously used in other domains (Yue & Li, 2024).

Neural Networks (GNNs) model such spatial dependencies (Jin et al., 2024). Methods such as MTGNN (Wu et al., 2020) and AGCRN (Bai et al., 2020) treat sensors as nodes and learn adjacency matrices to capture latent graph structure, while recent works combine temporal convolutions with graph attention to model spatiotemporal dynamics (Yi et al., 2023; Liu et al., 2024a). However, many approaches rely on predefined anatomical graphs, which may not reflect dynamic functional connectivity (Luo et al., 2024), or use fully connected graphs that are expensive and prone to overfitting (Fan et al., 2025). In contrast, MedMamba introduces a graph mamba module that learns an adaptive, data-driven dependency matrix, and imposes sparsity and acyclicity constraints for interpretability and efficiency, enabling discovery of complex channel relationships without prior anatomical knowledge.

# 3. Preliminaries

## 3.1. Subject Settings

MedTS are typically collected from multiple subjects and may include hierarchical levels such as subject, session, trial, and sample (Wang et al., 2023). In practice, long recordings are segmented into shorter samples for training, and each sample inherits a medical label (e.g., disease category) and a subject identifier. Evaluation protocol is therefore crucial, as different split strategies can lead to different conclusions. Figure 2 illustrates these two setups.

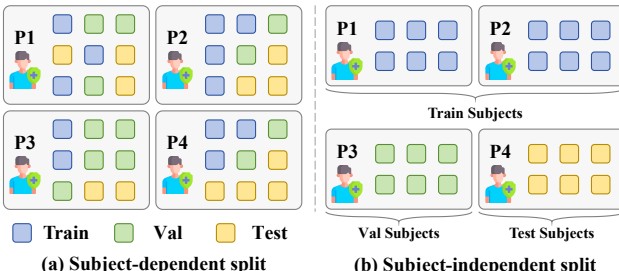

Figure 2. Illustration of subject-dependent and subject-independent evaluation. (a) Subject-dependent splits partition samples, allowing samples from the same subject to appear in both training and testing. (b) Subject-independent splits partition subjects, ensuring that all samples from a subject belong to a single split.

**Subject-dependent (SD).** In the SD setup, the dataset is split at the sample level, so samples from the same subject may appear in both training and test sets. This can introduce information leakage, since subject-specific characteristics (e.g., sensor placement and physiological baselines) may be shared across splits. SD evaluation is useful when deployment involves repeated measurements of the same individual or subject-specific calibration, but it can overestimate generalization to unseen subjects.

**Subject-independent (SI).** In the SI setup, the dataset is split at the subject level, assigning all samples from each subject to exactly one subset and ensuring disjoint training and testing subjects. This protocol better reflects clinical deployment, where models are applied to previously unseen individuals, and is generally more challenging due to cross-subject variability and distribution shifts. These shifts also amplify the impact of nonstationarity such as baseline drift. In this work, we report results under both setups and emphasize SI performance as the primary indicator of clinical generalization.

## 3.2. Problem Formulation

Consider a cohort of $N$ subjects denoted by $\{P_1, \ldots, P_N\}$. Each subject $P_n$ provides segmented time series samples $\{X_{P_n}^1, \ldots, X_{P_n}^{k_n}\}$, where $k_n$ is the number of samples for subject $P_n$. Each sample is a multichannel medical time series $X_{P_n}^i \in \mathbb{R}^{T \times C}$ with $T$ time steps and $C$ channels, associated with a class label $y_{P_n}^i \in \{1, \ldots, K\}$ indicating a disease or condition.

The goal is to learn a classifier $f(\cdot; \theta)$ that predicts $\hat{y}_{P_n}^i = f(X_{P_n}^i; \theta)$ and generalizes under either the SD or SI protocol described above. In MedMamba, $f$ is an end-to-end model that jointly captures long-range temporal dynamics and cross-channel dependencies while improving robustness to nonstationarities such as baseline drift.

# 4. Methodology

To address multi-scale temporal dynamics, nonstationarity (e.g., baseline drift), and latent cross-channel interactions in MedTS, we propose MedMamba. As illustrated in Figure 3, MedMamba combines an efficient state space backbone for long-range temporal modeling with medical-domain inductive biases. Given an input sample $\mathbf{X} \in \mathbb{R}^{T \times C}$, we first apply a **Multi-Scale Convolutional Embedding (MCE)** module to capture local morphology across multiple receptive fields, producing channel-wise embeddings in $\mathbb{R}^{T \times C \times D}$. Each subsequent layer contains (i) a **Tri-branch Differential State Space Encoder (TDSSE)** to mitigate drift via multi-view temporal modeling, and (ii) a **Spatial Graph Mamba (SGM)** module that learns a sample-conditioned directed dependency structure with sparsity and acyclicity priors. Stacking $L$ layers yields robust spatiotemporal representations for classification.

## 4.1. Multi-Scale Convolutional Embedding

Physiological waveforms exhibit discriminative patterns at multiple temporal scales, such as sharp spikes and sustained rhythms. A single linear projection is often insufficient to capture such local morphology (Wang et al., 2024). We therefore employ multi-scale temporal convolutions while keeping channel-wise tokens. Let $\mathcal{K} = \{k_1, \ldots, k_M\}$ be

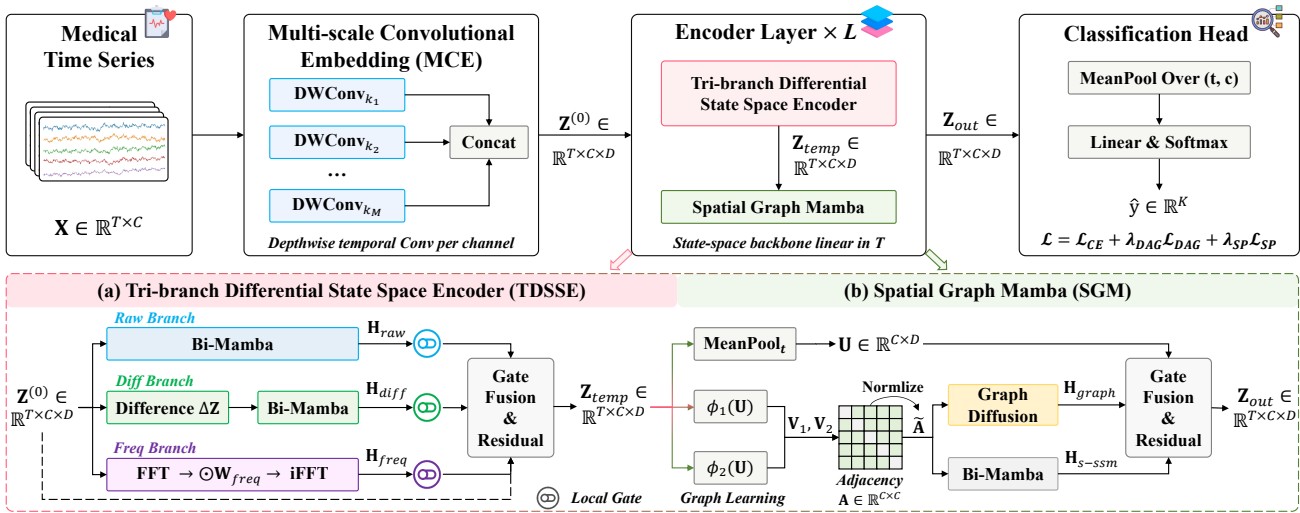

*Figure 3.* Overview of MedMamba architecture. The model consists of a MCE module followed by $L$ stacked encoder layers. Each layer integrates (a) **TDSSE** for multi-view temporal modeling and (b) **SGM** module for adaptive cross-channel dependency learning with sparsity and acyclicity priors. The final prediction is obtained by pooling and a linear classifier.

temporal kernel sizes. For each scale, we apply a depthwise temporal convolution per channel to get $\mathbf{E}^{(m)} \in \mathbb{R}^{T \times C \times d_m}$:

$$\mathbf{E}^{(m)} = \text{DWConv1D}_{k_m}(\mathbf{X}), \quad m = 1, \dots, M. \quad (1)$$

We concatenate features and project to width $D$:

$$\mathbf{Z}^{(0)} = \text{LN}(\text{Linear}(\text{Concat}[\mathbf{E}^{(1)}, \dots, \mathbf{E}^{(M)}])), \quad (2)$$

then $\mathbf{Z}^{(0)} \in \mathbb{R}^{T \times C \times D}$ is fed into the encoder stack.

### 4.2. Tri-Branch Differential State Space Encoder

MedTS are inherently non-stationary and prone to baseline drift. Relying solely on raw signal modeling can lead to representations that are fragile to such artifacts (Liu et al., 2023). To improve robustness, TDSSE builds three complementary temporal views: raw, first-order difference, and frequency-domain. Given $\mathbf{Z} \in \mathbb{R}^{T \times C \times D}$ (we omit the layer index for clarity), we process these views in parallel and fuse them via learnable gates.

**Raw View.** We use a bidirectional state space block along the time axis to capture long-range context efficiently (Gu & Dao, 2024). Let $\Psi(\cdot)$ denote the bidirectional Mamba operator applied to each channel sequence in parallel:

$$\mathbf{H}_{raw} = \Psi_{raw}(\mathbf{Z}) \in \mathbb{R}^{T \times C \times D}. \quad (3)$$

**Differential View.** To suppress slow-varying offsets and emphasize dynamic changes, we construct a first-order difference view. To avoid an artificial spike at the first step, we pad with zeros:

$$\Delta\mathbf{Z} = [\mathbf{0}; \mathbf{Z}_{2:T} - \mathbf{Z}_{1:T-1}] \in \mathbb{R}^{T \times C \times D}, \quad (4)$$

$$\mathbf{H}_{diff} = \Psi_{diff}(\Delta\mathbf{Z}) \in \mathbb{R}^{T \times C \times D}. \quad (5)$$

**Frequency View.** Spectral cues are critical for many physiological signals (e.g., EEG rhythms), and frequency-domain modeling can help isolate informative bands while attenuating artifacts (Zhou et al., 2022). We implement a learnable spectral modulation along the time axis using real FFT to ensure real-valued reconstruction:

$$\widehat{\mathbf{Z}} = \text{FFT}(\mathbf{Z}) \in \mathbb{C}^{F \times C \times D},$$
$$\widehat{\mathbf{H}}_{freq} = \widehat{\mathbf{Z}} \odot \mathbf{W}_{freq}, \quad (6)$$
$$\mathbf{H}_{freq} = \text{iFFT}(\widehat{\mathbf{H}}_{freq}) \in \mathbb{R}^{T \times C \times D},$$

where $F = \lfloor T/2 \rfloor + 1$ is the number of positive-frequency bins, $\odot$ denotes element-wise multiplication with broadcasting, and $\mathbf{W}_{freq} \in \mathbb{C}^{F \times 1 \times D}$ is a learnable complex filter.

**Gated Fusion.** The three views are complementary: raw preserves morphology, differential reduces sensitivity to drift, and frequency highlights rhythmic structure. For each branch $i \in \{raw, diff, freq\}$, we fuse them with two levels of gating. First, local gates for element-wise denoising:

$$\mathbf{g}_i = \sigma(\text{Linear}_i(\mathbf{H}_i)) \in \mathbb{R}^{T \times C \times D}, \widetilde{\mathbf{H}}_i = \mathbf{g}_i \odot \mathbf{H}_i. \quad (7)$$

Then, global gates for view selection:

$$\mathbf{s}_i = \text{MeanPool}_{t,c}(\widetilde{\mathbf{H}}_i) \in \mathbb{R}^D, \quad (8)$$

$$\boldsymbol{\alpha} = \text{Softmax}(\text{MLP}([\mathbf{s}_{raw}; \mathbf{s}_{diff}; \mathbf{s}_{freq}])) \in \mathbb{R}^3, \quad (9)$$

where $\alpha_i$ is broadcast to $\mathbb{R}^{T \times C \times D}$. The fused temporal representation is

$$\mathbf{Z}_{temp} = \sum_i \alpha_i \widetilde{\mathbf{H}}_i + \mathbf{Z} \in \mathbb{R}^{T \times C \times D}. \quad (10)$$

## 4.3. Spatial Graph Mamba Module

Multichannel MedTS contain diagnostic cues in cross-channel interactions, yet predefined anatomical graphs may be unavailable or mismatched to sample-specific functional dependencies. To model such interactions in a data-driven manner, we propose SGM, which performs sample-conditioned adaptive graph learning and integrates it with a channel-wise state space pathway for spatial modeling. For each input sample, SGM learns a directed dependency matrix and regularizes it towards sparsity and acyclicity for robustness and interpretability.

**Sample-Conditioned Graph Learning.** Given temporal features $\mathbf{Z}_{temp}$, we first summarize each channel by temporal pooling to obtain a compact node representation:

$$\mathbf{u}_c = \text{MeanPool}_t\big(\mathbf{Z}_{temp}[:,c,:]\big) \in \mathbb{R}^D, \quad (11)$$

$$\mathbf{U} = [\mathbf{u}_1;\ldots;\mathbf{u}_C] \in \mathbb{R}^{C\times D}. \quad (12)$$

We then produce sample-dependent node embeddings via two learnable projections $\phi_1, \phi_2$:

$$\mathbf{V}_1 = \phi_1(\mathbf{U}),\ \mathbf{V}_2 = \phi_2(\mathbf{U}) \in \mathbb{R}^{C\times d_{node}}, \quad (13)$$

A directed edge weight from node $j$ to node $i$ is produced by a bilinear affinity with self-loop masking:

$$\mathbf{A} = \sigma\big(\mathbf{V}_1\mathbf{V}_2^\top\big) \odot (\mathbf{1} - \mathbf{I}) \in \mathbb{R}^{C\times C}, \quad (14)$$

where $\sigma(\cdot)$ is the sigmoid function and $\mathbf{I}$ is the identity matrix. Importantly, $\mathbf{A}$ is computed for each sample, enabling adaptive graph learning that captures sample-specific channel dependencies. To perform message diffusion, we construct a row-stochastic transition matrix via random-walk normalization:

$$\mathbf{d} = \mathbf{A}\mathbf{1} \in \mathbb{R}^C, \qquad \widetilde{\mathbf{A}}_{ij} = \frac{\mathbf{A}_{ij}}{\mathbf{d}_i + \varepsilon}, \quad (15)$$

where $\mathbf{1}$ is an all-ones vector and $\varepsilon > 0$ avoids division by zero. Compared with row-wise softmax, degree normalization preserves the near-zero structure induced by sparsity priors, making the learned graph consistent with the structure regularization below.

**Structure Priors.** Dense dependency graphs may overfit and are difficult to interpret. We therefore impose two structure priors on the *pre-normalized* adjacency $\mathbf{A}$:

$$\mathcal{L}_{SP} = \|\mathbf{A}\|_1, \qquad \mathcal{L}_{DAG} = \text{tr}\big(\exp(\mathbf{A}\odot\mathbf{A})\big) - C, \quad (16)$$

where $\|\cdot\|_1$ is the element-wise $\ell_1$ norm and $\exp(\cdot)$ denotes the matrix exponential. $\mathcal{L}_{SP}$ encourages sparse dependencies, while $\mathcal{L}_{DAG}$ promotes acyclicity to improve stability and interpretability. We provide detailed explanation and proof of adaptive graph learning in the Appendix B.

**Spatial Fusion.** SGM combines two complementary spatial pathways. The graph diffusion pathway propagates information along the learned directed transition matrix $\widetilde{\mathbf{A}}$, capturing structured local interactions. In parallel, the channel-wise SSM pathway treats the channel axis as a sequence and models global spatial dependencies with linear-time state space modeling. Let $\mathbf{Z}_{temp}^{(t)} \in \mathbb{R}^{C\times D}$ denote the channel-token matrix at time step $t$. The graph diffusion is:

$$\mathbf{H}_{graph}^{(t)} = \mathbf{Z}_{temp}^{(t)}\mathbf{W}_0 + \widetilde{\mathbf{A}}\,\mathbf{Z}_{temp}^{(t)}\mathbf{W}_1 \in \mathbb{R}^{C\times D}, \quad (17)$$

with learnable $\mathbf{W}_0, \mathbf{W}_1 \in \mathbb{R}^{D\times D}$. For the channel-wise SSM pathway, we apply a bidirectional SSM over channels at each $t$:

$$\mathbf{H}_{s\text{-}ssm}^{(t)} = \Psi_{spatial}\big(\widetilde{\mathbf{A}}\mathbf{Z}_{temp}^{(t)}\big) \in \mathbb{R}^{C\times D}. \quad (18)$$

We then fuse the two pathways using an element-wise gate:

$$\boldsymbol{\lambda}^{(t)} = \sigma(\text{Linear}(\mathbf{H}_{graph}^{(t)}) + \text{Linear}(\mathbf{H}_{s\text{-}ssm}^{(t)})), \quad (19)$$

$$\mathbf{Z}_{out}^{(t)} = \text{Linear}(\boldsymbol{\lambda}^{(t)}\odot\mathbf{H}_{graph}^{(t)} + (1-\boldsymbol{\lambda}^{(t)})\odot\mathbf{H}_{s\text{-}ssm}^{(t)}) + \mathbf{Z}_{temp}^{(t)}. \quad (20)$$

Stacking $\{\mathbf{Z}_{out}^{(t)}\}_{t=1}^T$ yields $\mathbf{Z}_{out} \in \mathbb{R}^{T\times C\times D}$, which is passed to the next layer or the classifier.

## 4.4. Prediction and Training Objective

We predict the label by pooling $\mathbf{Z}_{out} \in \mathbb{R}^{T\times C\times D}$ and applying a linear classifier:

$$\hat{\mathbf{y}} = \text{Softmax}(\text{Linear}(\text{MeanPool}_{t,c}(\mathbf{Z}_{out}))) \in \mathbb{R}^K. \quad (21)$$

MedMamba is trained end-to-end with cross-entropy loss, augmented with graph structure regularization:

$$\mathcal{L} = \mathcal{L}_{CE} + \lambda_{SP}\mathcal{L}_{SP} + \lambda_{DAG}\mathcal{L}_{DAG}, \quad (22)$$

where $\lambda_{DAG}$ and $\lambda_{SP}$ control the strengths of the acyclicity and sparsity priors, respectively.

## 5. Experiments

### 5.1. Experimental Settings

**Datasets.** We conduct empirical analyses on five representative medical datasets, i.e., ADFTD (Miltiadous et al., 2023b), APAVA (Escudero et al., 2006), TDBRAIN (Van Dijk et al., 2022), PTB (PhysioBank, 2000), and PTB-XL (Wagner et al., 2020). These datasets include three EEG datasets and two ECG datasets. The data preprocessing and split are following the previous work (Wang et al., 2024). For further details, please refer to the Appendix A.1.

**Baselines.** The baseline methods used for comparison include six SOTA time series analysis approaches: GPT4TS

*Table 1.* **Classification results of our MedMamba and baseline models on the APAVA dataset under the subject-dependent setting**. The best results are highlighted in **red**, and the second-best in **blue**. Note that the percentage symbol is omitted.

| Dataset | Metric | Ours MedMamba | WWW'25 MedGNN | ACM MM'25 KEMed | Neurips'24 MedFormer | ICLR'24 iTransformer | Neurips'24 FourierGNN | InfoS'24 TodyNet | IJCAI'24 MTST | ICLR'23 PatchTST | Neurips'23 GPT4TS |
|---|---|---|---|---|---|---|---|---|---|---|---|
| APAVA | Accuracy | **98.86**±0.09 | 98.16±0.54 | **98.29**±0.28 | 96.26±0.34 | 91.34±0.37 | 67.09±1.27 | 96.63±0.43 | 75.06±0.65 | 94.22±0.70 | 98.09±0.32 |
| | Precision | **98.77**±0.11 | 98.16±0.59 | **98.27**±0.32 | 96.88±0.30 | 90.99±0.26 | 65.58±1.41 | 96.82±0.40 | 73.83±0.76 | 94.08±0.77 | 98.14±0.16 |
| | Recall | **98.87**±0.09 | 98.02±0.58 | **98.17**±0.30 | 95.44±0.43 | 91.03±0.71 | 64.97±0.99 | 96.73±0.50 | 72.69±0.60 | 93.91±0.79 | 97.89±0.50 |
| | F1 Score | **98.81**±0.10 | 98.08±0.56 | **98.22**±0.29 | 96.05±0.37 | 90.97±0.44 | 65.12±1.02 | 96.78±0.45 | 73.01±0.54 | 93.86±0.73 | 98.00±0.34 |
| | AUROC | **99.96**±0.01 | 99.84±0.08 | **99.85**±0.04 | 99.69±0.10 | 97.04±0.08 | 71.36±1.50 | 99.53±0.04 | 91.33±0.52 | 98.83±0.24 | 99.81±0.01 |
| | AUPRC | 99.82±0.03 | **99.83**±0.09 | 99.82±0.06 | 99.70±0.10 | 96.49±0.13 | 69.77±1.69 | 99.53±0.04 | 79.32±0.94 | 98.77±0.26 | **99.83**±0.01 |

*Table 2.* **Classification results of our MedMamba and baseline models on the 5 different datasets under the subject-independent setting.** The best results are highlighted in **red**, and the second-best in **blue**. Note that the percentage symbol is omitted.

| Dataset | Metric | Ours MedMamba | WWW'25 MedGNN | ACM MM'25 KEMed | Neurips'24 MedFormer | ICLR'24 iTransformer | Neurips'24 FourierGNN | InfoS'24 TodyNet | IJCAI'24 MTST | ICLR'23 PatchTST | Neurips'23 GPT4TS |
|---|---|---|---|---|---|---|---|---|---|---|---|
| ADFTD | Accuracy | **57.56**±0.93 | **56.12**±0.11 | 51.85±2.92 | 53.27±1.54 | 52.60±1.59 | 49.95±0.92 | 45.18±2.29 | 45.60±2.03 | 43.97±1.63 | 54.11±0.67 |
| | Precision | **56.01**±0.81 | **55.07**±0.09 | 48.75±1.11 | 51.02±1.57 | 46.78±1.27 | 43.15±1.57 | 43.82±2.50 | 44.70±1.33 | 42.14±1.46 | 47.76±0.94 |
| | Recall | **57.41**±0.79 | **55.47**±0.34 | 48.25±1.37 | 50.71±1.55 | 47.27±1.29 | 44.51±1.17 | 42.48±2.24 | 45.05±1.30 | 41.36±0.85 | 46.63±0.87 |
| | F1 Score | **55.98**±0.82 | **55.00**±0.34 | 47.66±1.10 | 50.65±1.51 | 46.79±1.13 | 43.59±1.33 | 42.22±2.73 | 44.31±1.74 | 41.26±1.16 | 44.66±0.58 |
| | AUROC | **75.72**±0.59 | **74.68**±0.33 | 68.19±0.92 | 70.93±1.19 | 67.26±1.16 | 62.39±1.49 | 61.08±2.64 | 65.20±0.81 | 59.29±1.04 | 67.86±0.52 |
| | AUPRC | **58.85**±0.67 | **57.51**±0.38 | 51.06±1.30 | 51.21±1.32 | 49.53±1.21 | 45.60±1.18 | 43.82±2.35 | 45.16±0.85 | 41.49±1.32 | 49.97±0.67 |
| APAVA | Accuracy | **84.48**±0.55 | **82.60**±0.35 | 78.74±1.48 | 78.74±0.64 | 74.55±1.66 | 57.25±2.05 | 73.05±4.57 | 71.14±1.59 | 67.38±1.83 | 77.78±4.23 |
| | Precision | **86.79**±0.62 | **87.70**±0.22 | 83.17±2.06 | 81.11±0.84 | 74.77±2.10 | 55.03±1.07 | 75.26±7.06 | 79.30±0.97 | 78.31±1.59 | 79.36±5.67 |
| | Recall | **81.87**±0.27 | **78.93**±0.09 | 72.15±2.22 | 75.40±0.66 | 71.76±1.72 | 54.55±1.42 | 71.06±2.86 | 65.27±2.28 | 60.41±2.30 | 75.36±3.22 |
| | F1 Score | **83.03**±0.49 | **80.25**±0.16 | 72.58±2.58 | 76.31±0.71 | 72.30±1.79 | 54.27±1.51 | 71.05±3.34 | 64.01±3.16 | 56.81±3.60 | 76.00±3.71 |
| | AUROC | 86.30±0.17 | 85.93±0.26 | **88.54**±1.18 | 83.20±0.91 | 85.59±1.55 | 56.43±2.13 | 77.52±5.97 | 68.87±2.34 | 65.20±4.30 | 85.55±4.64 |
| | AUPRC | 87.07±0.99 | **87.33**±0.82 | **88.68**±1.09 | 83.66±0.92 | 84.39±1.57 | 55.21±1.72 | 78.44±5.24 | 71.06±1.60 | 67.61±3.76 | 85.62±5.09 |
| PTB | Accuracy | **89.36**±0.45 | **84.53**±0.28 | 82.47±1.35 | 83.50±2.01 | 81.89±0.74 | 76.02±2.99 | 75.98±2.73 | 76.59±1.90 | 77.19±1.53 | 82.31±0.94 |
| | Precision | **90.01**±0.48 | **87.35**±0.45 | 86.57±2.32 | 85.19±0.94 | 83.26±1.26 | 79.42±3.84 | 79.24±3.48 | 79.88±1.90 | 79.40±1.18 | 84.73±0.81 |
| | Recall | **85.70**±1.02 | **77.90**±0.66 | 74.51±1.53 | 77.11±3.39 | 73.39±1.02 | 65.40±4.57 | 65.57±4.29 | 66.31±2.95 | 67.68±2.49 | 73.04±1.58 |
| | F1 Score | **87.36**±0.96 | **80.40**±0.62 | 77.01±1.78 | 79.18±3.31 | 75.06±1.08 | 66.04±5.94 | 66.23±5.75 | 67.38±3.71 | 69.06±2.87 | 73.31±1.53 |
| | AUROC | 90.13±0.93 | **93.31**±0.46 | 91.71±2.38 | **92.81**±1.48 | 91.57±0.98 | 84.93±3.56 | 90.27±2.50 | 86.86±2.75 | 88.48±2.02 | 90.05±0.58 |
| | AUPRC | 88.73±1.11 | 87.74±2.54 | **90.95**±2.41 | 90.32±1.54 | 89.41±0.78 | 82.03±4.30 | 87.16±3.24 | 83.75±2.84 | 83.54±1.92 | **90.87**±1.49 |
| PTB-XL | Accuracy | **73.86**±0.30 | **73.87**±0.18 | 73.77±0.41 | 72.87±0.23 | 69.35±0.15 | 64.38±0.25 | 73.41±0.45 | 72.14±0.27 | 73.15±0.24 | 72.92±0.26 |
| | Precision | 66.15±0.67 | **66.26**±0.29 | **66.30**±0.90 | 64.14±0.42 | 59.73±0.24 | 52.23±0.29 | 66.22±0.76 | 63.84±0.72 | 65.66±0.53 | 65.15±0.41 |
| | Recall | **61.30**±1.08 | 61.13±0.23 | **62.17**±0.49 | 60.60±0.46 | 54.64±0.19 | 49.22±0.74 | 58.96±0.81 | 60.01±0.81 | 60.51±0.12 | 60.19±0.63 |
| | F1 Score | **63.20**±0.91 | 62.54±0.20 | **63.09**±0.12 | 62.02±0.37 | 56.25±0.14 | 49.99±0.62 | 60.84±0.85 | 61.43±0.38 | 62.35±0.22 | 62.12±0.27 |
| | AUROC | 89.33±0.22 | **90.21**±0.15 | **89.86**±0.17 | 89.66±0.13 | 86.75±0.06 | 82.46±0.21 | 89.00±0.39 | 88.97±0.33 | 89.62±0.21 | 89.64±0.10 |
| | AUPRC | 66.91±0.75 | 66.01±0.88 | **67.24**±0.32 | 66.39±0.22 | 60.36±0.09 | 52.81±0.48 | 66.28±0.76 | 65.83±0.51 | **67.22**±0.29 | 66.61±0.16 |
| TDBRAIN | Accuracy | **92.13**±0.41 | 91.04±0.09 | 87.29±2.77 | 89.62±0.81 | 74.73±1.05 | 67.10±1.40 | 86.85±2.31 | 76.96±3.76 | 78.54±5.16 | 86.90±2.35 |
| | Precision | **92.33**±0.37 | 91.15±0.12 | 87.46±2.65 | 89.68±0.78 | 74.78±1.04 | 67.18±1.45 | 87.04±2.17 | 77.24±3.59 | 78.76±5.31 | 86.91±2.33 |
| | Recall | **92.13**±0.41 | 91.04±0.20 | 87.29±2.77 | 89.62±0.81 | 74.73±1.05 | 67.10±1.40 | 86.85±2.31 | 76.96±3.76 | 78.54±5.16 | 86.90±2.35 |
| | F1 Score | **92.12**±0.42 | 91.04±0.20 | 87.28±2.78 | 89.62±0.81 | 74.72±1.05 | 67.07±1.37 | 86.85±2.31 | 76.88±3.83 | 78.51±5.15 | 86.89±2.35 |
| | AUROC | 95.62±0.66 | **96.74**±0.08 | 95.25±2.37 | **96.41**±0.35 | 83.38±1.14 | 73.21±1.50 | 89.79±0.33 | 85.27±4.46 | 86.17±5.46 | 94.55±1.50 |
| | AUPRC | **96.68**±0.63 | 95.35±0.35 | 95.39±2.44 | **96.51**±0.33 | 83.76±1.29 | 71.40±1.44 | 94.80±0.31 | 82.81±5.64 | 83.75±7.23 | 94.71±1.46 |

(Zhou et al., 2023), MTST (Zhang et al., 2024), FourierGNN (Yi et al., 2023), iTransformer (Liu et al., 2024b), PatchTST (Nie et al., 2023), TodyNet (Liu et al., 2024a), with three methods specifically designed for MedTS, Medformer (Wang et al., 2024), MedGNN (Fan et al., 2025) and KEMed (Yuan et al., 2025) [2].

**Implementation.** We employ six evaluation metrics: accuracy, precision, recall, F1 score, AUROC, and AUPRC. All models are trained end-to-end under the same SD/SI protocols using the fixed training, validation, and test splits, and hyperparameters are selected based on validation performance. For MedMamba, we optimize the classification objective together with the graph-structure regularizers in Eq. (22). To reduce randomness, we repeat each experiment with five different random seeds and report the mean and standard deviation. All experiments are run on 4 GeForce

RTX 4090 GPUs. Detailed training configurations (optimizer, learning rate schedule, batch size, epochs/early stopping, and regularization) are provided in the Appendix A.2.

**5.2. Main Results**

**Subject-Dependent Evaluation.** In the SD setting, multiple samples from the same subject can appear in the training, validation, and test sets, reflecting clinical scenarios where patients may have repeated visits and the model makes multiple predictions for the same individual. Table 1 compares MedMamba with several strong baselines on the APAVA dataset. MedMamba achieves the best performance on five metrics. For AUPRC, MedMamba remains highly competitive (99.82%) and ranks among the top methods. These results indicate that APAVA contains highly discriminative Alzheimer-related patterns that most competitive models can capture (e.g., KEMed, MedGNN), all achieving strong

---

[2]Specific comparison with Mamba baseline in Appendix E.8.

*Table 3.* **Component ablation under the SI setting on ADFTD and PTB.** We report mean±std. ↓ △ indicates the drop relative to the full MedMamba. "**w/o MCE**" replaces multi-scale convolutions with a single linear embedding; "**w/o TDSSE**" keeps only the raw-view bidirectional Mamba branch; "**w/o SGM**" removes graph diffusion and retains only the channel-wise SSM pathway.

| Dataset | Metric | Full | w/o MCE | w/o TDSSE | w/o SGM |
|---------|--------|------|---------|-----------|---------|
| **ADFTD** | Accuracy | **57.56±0.93** | 56.41±0.98 (↓1.15) | 53.22±1.06 (↓4.34) | 54.47±1.01 (↓3.09) |
| | Precision | **56.01±0.81** | 54.93±0.87 (↓1.08) | 51.88±0.94 (↓4.13) | 53.16±0.89 (↓2.85) |
| | Recall | **57.41±0.79** | 56.18±0.86 (↓1.23) | 53.04±0.92 (↓4.37) | 54.02±0.87 (↓3.39) |
| | F1 Score | **55.98±0.82** | 54.86±0.90 (↓1.12) | 51.69±0.96 (↓4.29) | 52.95±0.89 (↓3.03) |
| | AUROC | **75.72±0.59** | 74.48±0.65 (↓1.24) | 71.76±0.72 (↓3.96) | 72.85±0.67 (↓2.87) |
| | AUPRC | **58.85±0.67** | 57.73±0.74 (↓1.12) | 54.32±0.81 (↓4.53) | 55.88±0.76 (↓2.97) |
| **PTB** | Accuracy | **89.36±0.45** | 87.29±0.53 (↓2.07) | 84.11±0.56 (↓5.25) | 84.74±0.60 (↓4.62) |
| | Precision | **90.01±0.48** | 88.18±0.57 (↓1.83) | 84.97±0.59 (↓5.04) | 85.59±0.62 (↓4.42) |
| | Recall | **85.70±1.02** | 83.42±1.11 (↓2.28) | 80.21±1.14 (↓5.49) | 80.95±1.17 (↓4.75) |
| | F1 Score | **87.36±0.96** | 85.18±1.06 (↓2.18) | 81.96±1.09 (↓5.40) | 82.57±1.03 (↓4.79) |
| | AUROC | **90.13±0.93** | 88.27±0.99 (↓1.86) | 85.01±1.01 (↓5.12) | 85.68±0.97 (↓4.45) |
| | AUPRC | **88.73±1.11** | 86.91±1.17 (↓1.82) | 83.60±1.20 (↓5.13) | 84.04±1.22 (↓4.69) |

performance under this easier protocol. However, SD evaluation may overestimate true generalization due to potential information leakage across splits, as discussed in Section 3.1. We also provide more results in Appendix E.1.

**Subject-Independent Evaluation.** To more objectively assess clinical generalization to unseen subjects, we further conduct experiments under the SI setting, in which the training, validation, and test sets are split by subjects. Each subject, together with all corresponding samples, is assigned to exactly one split according to a predefined ratio or subject IDs, ensuring that samples from the same subject never appear in multiple sets. This protocol prevents subject overlap and provides a more objective assessment of clinical generalization. Table 2 reports the SI results on five datasets. Overall, MedMamba achieves leading performance on most datasets, obtaining 20 top-1 and 3 top-2 results out of 30 entries. Notably, MedMamba ranks first in terms of F1 score across all five datasets, demonstrating a strong balance between precision and recall under distribution shifts across subjects. Such a balance is particularly important in medical diagnosis, where both false positives and false negatives can lead to serious consequences. More detailed analysis can be found in the Appendix E.2.

### 5.3. Ablation Studies

**Component Ablation.** We perform component ablations under the SI setting by removing MCE, TDSSE, or SGM while keeping all other parts unchanged. Table 3 shows results on ADFTD and PTB. Overall, the full MedMamba achieves the best performance across metrics on both datasets, indicating that each component is beneficial. On ADFTD, removing TDSSE causes the largest drop, highlighting the importance of multi-view temporal modeling for nonstationarity. On PTB, removing either TDSSE or SGM leads to pronounced degradations, showing that robust temporal modeling and adaptive cross-channel dependency learning are both critical. Removing MCE yields smaller but consistent decreases, suggesting multi-scale morphology

encoding complements the temporal and spatial modules. We provide full-dataset ablations in the Appendix E.3.

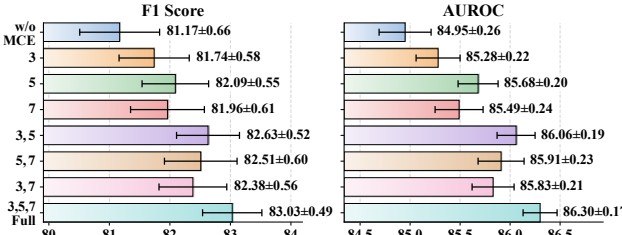

*Figure 4.* **MCE ablation on APAVA under the SI setting.** Performance of different convolutional kernel configurations in MCE.

**MCE Ablation.** Figure 4 shows the performance of MCE on APAVA. Replacing MCE with a single linear embedding yields the lowest scores, while using a single kernel already helps, with $k=5$ outperforming $k=3$ and $k=7$ on both metrics. Combining two scales further improves results, where $(3, 5)$ is the strongest two-scale variant. The full design $(3, 5, 7)$ achieves the best overall performance, confirming that multi-scale encoding provides complementary benefits beyond any single receptive field. Results on all datasets are reported in the Appendix E.4.

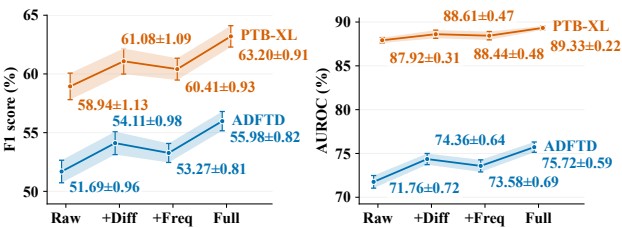

*Figure 5.* **Tri-branch ablation results under the SI setting on ADFTD and PTB-XL.** Left is F1 Score and right is AUROC.

**Tri-Branch Ablation.** Figure 5 shows that adding either the differential or frequency branch consistently improves over the raw-only model on both ADFTD and PTB-XL, and the full tri-branch design achieves the best F1 and AUROC. The gains are more pronounced on ADFTD, indicating that multi-view temporal modeling is particularly beneficial under stronger nonstationarity. Full results on all datasets are provided in the Appendix E.5.

**SGM Ablation.** Figure 6 demonstrates that both adaptive graph learning and structure priors contribute to robust subject-independent performance. Replacing the learned, sample-conditioned dependency graph with a fixed graph leads to the most noticeable degradation on APAVA and TDBRAIN, indicating that modeling sample-specific cross-channel interactions is important. Moreover, removing either the sparsity prior or the DAG-inspired regularizer consistently reduces performance, suggesting that these priors help prevent overly dense or unstable dependency patterns. Full results on all datasets are provided in the Appendix E.6.

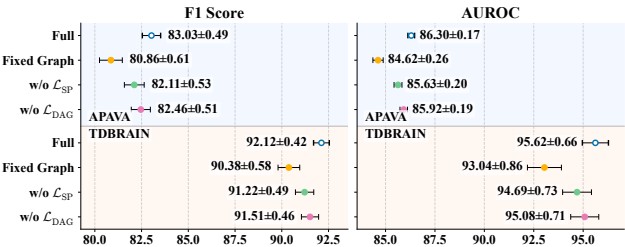

Figure 6. **SGM ablation on APAVA and TDBRAIN under the SI setting.** We compare the **Full** model with three variants: **"Fixed Graph"** (no learning), **"w/o $\lambda_{SP}$"** (sparsity prior), and **"w/o $\lambda_{DAG}$"** (DAG prior).

### 5.4. Additional Experiments

**Robustness to baseline drift.** We implement drift by adding a per-channel linear baseline to each test sample: $X'_{t,c} = X_{t,c} + s \cdot a_c \cdot (\frac{t}{T} - \frac{1}{2})$, where $s \in \{0, 0.25, 0.5, 0.75, 1.0\}$ and $a_c \sim \text{Uniform}(-\sigma_c, \sigma_c)$ is sampled independently for each channel (with $\sigma_c$ the within-sample std of channel $c$). Figure 7 shows that MedMamba degrades more gracefully as drift strength increases on APAVA, maintaining consistently higher F1 and AUROC than both the raw-only variant (w/o TDSSE) and the graph baseline MedGNN. Removing TDSSE leads to a much steeper performance drop under strong drift, indicating that the proposed differential and frequency-aware views are crucial for mitigating low-frequency baseline shifts. These results support that MedMamba's multi-view temporal modeling improves robustness to nonstationary artifacts, which is important for reliable clinical deployment.

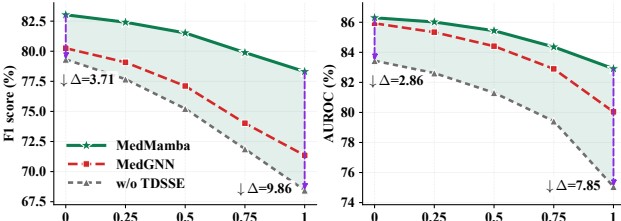

Figure 7. **Robustness to baseline drift on APAVA under the SI setting.** We inject controlled baseline drift (horizontal axis) with strength $\{0, 0.25, 0.5, 0.75, 1.0\}$ and compare MedMamba with **"w/o TDSSE"** (raw-only variant) and **MedGNN**. Left is F1 Score and right is AUROC.

**Efficiency Analysis.** Figures 8 compare accuracy against training time per epoch and peak GPU memory on APAVA and TDBRAIN, respectively. Overall, MedMamba achieves strong accuracy while maintaining a favorable efficiency profile, occupying the lower-memory and faster-training region compared with many recent Transformer- and graph-based baselines. This suggests that the SSM-based backbone together with lightweight multi-view and graph components provides a practical trade-off between predictive performance and computational cost for medical time series classification.

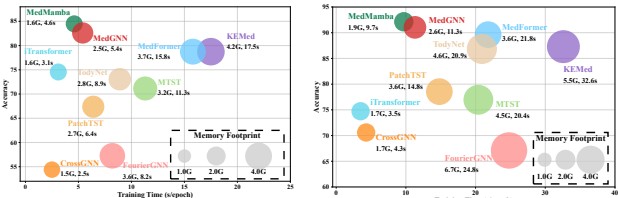

Figure 8. **Efficiency on APAVA (left) and TDBRAIN (right) under the SI setting.** Accuracy versus training time per epoch and peak GPU memory for MedMamba and baselines.

**Robustness to missing channels.** Figure 9 evaluates sensor-dropout robustness on PTB. For each test sample, we simulate sensor failure by uniformly sampling $\lfloor pC \rfloor$ channels without replacement and zeroing their trajectories ($X'_{:,c} = 0$). As the missing rate $p$ increases, all methods suffer performance degradation; however, MedMamba consistently outperforms both the fixed-graph variant and MedGNN. The widening gap between MedMamba and the fixed-graph baseline under severe channel loss underscores the importance of learning sample-adaptive dependencies for maintaining reliability given imperfect sensor availability. We also provide qualitative analysis of learned graphs in Appendix E.9.

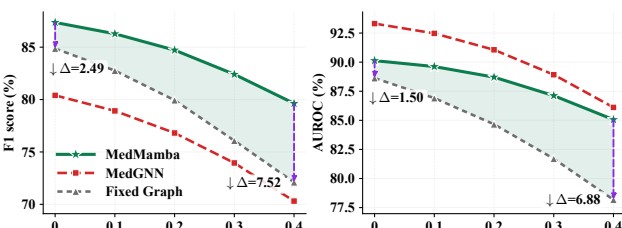

Figure 9. **Robustness to missing channels on PTB under the SI setting.** We randomly mask channels with missing rates $\{0, 0.1, 0.2, 0.3, 0.4\}$ at test time and compare MedMamba with **"Fixed Graph"** variant (no graph learning) and **MedGNN**. Left is F1 Score and right is AUROC.

## 6. Conclusion

In this paper, we proposed **MedMamba**, an end-to-end architecture for medical time series classification that combines state space modeling with medical-domain inductive biases. MedMamba integrates a multi-scale convolutional embedding for local morphology, a tri-branch differential state space encoder that fuses raw/difference/frequency views for nonstationarity robustness, and an adaptive spatial graph module that learns sample-conditioned cross-channel dependencies with sparsity and acyclicity regularization. Experiments on five real-world datasets under subject-dependent and subject-independent protocols show consistent gains over strong baselines, together with improved robustness to baseline drift and missing channels and competitive efficiency.

## Acknowledgment

This work was supported in part by grants from the National Natural Science Foundation of China (62576284 & 62306241), and in part by grants from the Innovation Foundation for Doctor Dissertation of Northwestern Polytechnical University (No.CX2025109).

## Impact Statement

This paper presents a multichannel medical time series framework that combines multi-view temporal encoding with adaptive dependency learning to better capture long-range dynamics and cross-channel interactions under nonstationary conditions. Our work may offer practical value for automatic analysis of physiological signals (e.g., EEG/ECG) in clinical decision support and may inspire future research on robust and efficient sequence modeling in healthcare. All datasets used in our experiments are publicly available and de-identified, and we do not access or release any private patient data. We have carefully considered the ethical implications of our work and do not foresee additional ethical or moral risks beyond those commonly associated with the use of machine learning models in medicine.

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

# A. Experimental Details

## A.1. Datasets

We conduct experiments on five public MedTS datasets, including three EEG datasets (ADFTD (Miltiadous et al., 2023b), APAVA (Escudero et al., 2006), TDBRAIN (Van Dijk et al., 2022)) and two ECG datasets (PTB (PhysioBank, 2000), and PTB-XL (Wagner et al., 2020)). All datasets are publicly available and contain de-identified physiological signals. We follow the established preprocessing and splitting protocols used in prior work for fair comparison (Wang et al., 2024).

**APAVA (EEG, 2-class).** APAVA contains 23 subjects (12 AD, 11 HC) with 16 channels. Each trial is a 5-second sequence with 1280 timestamps. We standardize each trial and segment it into 1-second samples (256 timestamps) with half-overlap, resulting in 5,967 samples. We employ a subject-independent split using predefined subject IDs (validation: $\{15,16,19,20\}$, test: $\{1,2,17,18\}$; remaining subjects for training).

**TDBRAIN (EEG, 2-class subset).** TDBRAIN is a large EEG dataset with 33 channels. Following prior work, we use an eye-closed subset containing 25 Parkinson's disease (PD) subjects and 25 healthy controls (HC). Each trial is segmented into non-overlapping 1-second samples (256 timestamps), discarding segments shorter than 1 second, yielding 6,240 samples. We use a subject-independent split with predefined subject IDs (validation: $\{18,19,20,21,46,47,48,49\}$, test: $\{22,23,24,25,50,51,52,53\}$; remaining subjects for training).

**ADFTD (EEG, 3-class).** ADFTD includes 36 AD, 23 FTD, and 29 HC subjects with 19 channels (raw sampling rate 500 Hz). We apply a 0.5–45 Hz bandpass filter, downsample to 256 Hz, and segment recordings into non-overlapping 1-second samples (256 timestamps), discarding segments shorter than 1 second. We report results under both subject-dependent (sample-level split) and subject-independent (subject-level split) protocols with a 60/20/20 ratio for train/validation/test.

**PTB (ECG, 2-class subset).** PTB is an ECG dataset with 15 channels (raw sampling rate 1000 Hz). We use a commonly adopted subset for binary classification (myocardial infarction vs. healthy control). We downsample to 250 Hz, normalize signals, extract single-heartbeat segments based on R-peak detection with outlier removal, and pad to a unified length, resulting in 64,356 samples. We employ a subject-independent split with a 60/20/20 ratio over subjects.

**PTB-XL (ECG, 5-class).** PTB-XL contains 12-channel 10-second ECG recordings with multi-label diagnoses. Following prior work, we discard subjects with inconsistent diagnoses across trials, keeping 17,596 subjects. We use the 500 Hz version, downsample to 250 Hz, normalize, and segment each trial into non-overlapping 1-second samples (250 timestamps), yielding 191,400 samples. We use a subject-independent split with a 60/20/20 ratio over subjects.

## A.2. Implementation Details

**General Setup.** All models were implemented in PyTorch and trained on a server equipped with four NVIDIA GeForce RTX 4090 GPUs. We utilized the Adam optimizer with a weight decay of $1 \times 10^{-5}$ and employed a cosine annealing scheduler to adjust the learning rate during the training process. The maximum number of epochs was set to 100, with an early stopping mechanism (patience of 10 epochs) monitoring the validation F1 score to prevent overfitting. To ensure reproducibility, all experiments were repeated with five distinct random seeds (41–45), and we report the mean and standard deviation. Regarding the general architecture of MedMamba, we consistently utilized depthwise separable convolutions with kernel scales of $\{3, 5, 7\}$ for the Multi-scale Convolutional Embedding (MCE) module and set the expansion factor $E = 2$ for the State Space Model backbone, using SiLU as the activation function. We provide pseudocode at Algorithm 1. For the settings of other baselines, we refer to their official implementations.

**Dataset-Specific Configurations.** To adapt MedMamba to the varying scales and characteristics of different medical datasets, we tailored the key hyperparameters—specifically the hidden dimension ($D$), number of layers ($L$), state dimension ($d_{state}$), batch size ($B$), and learning rate ($LR$)—as follows:

- **ADFTD:** We configured the model with a hidden dimension $D = 128$ and a depth of $L = 4$ layers. The SSM state dimension was set to $d_{state} = 16$ with a convolution width of 4. We used a batch size of 128 and a higher learning rate of $5 \times 10^{-4}$, applying a dropout of 0.2 and graph regularization $\lambda = 0.3$.

- **APAVA:** Given the complexity of this dataset, we increased the model capacity to $D = 256$ while keeping $L = 4$. The state dimension was doubled to $d_{state} = 32$. Training was conducted with a smaller batch size of $64$ and a learning rate of $1 \times 10^{-4}$.

- **PTB:** For this dataset, a shallower network sufficed. We set $D = 128$ and $L = 2$, with $d_{state} = 16$. The graph regularization was set stronger at $\lambda = 0.5$. We used a batch size of $128$, a learning rate of $1 \times 10^{-4}$, and a lower dropout of $0.1$.

- **PTB-XL:** Similar to PTB, we used $D = 128$ and $L = 2$, but with a reduced state dimension of $d_{state} = 8$ to prevent overfitting on the multi-class labels. The batch size was $128$ with a learning rate of $1 \times 10^{-4}$ and dropout of $0.2$.

- **TDBRAIN:** We utilized $D = 128$ and $L = 4$. Notably, for this dataset, we found that a smaller convolution width ($d_{conv} = 2$) yielded better results, and we omitted the explicit graph regularization ($\lambda = 0$). The model was trained with a large batch size of $256$ and a learning rate of $5 \times 10^{-4}$.

---

**Algorithm 1** Training Procedure of MedMamba

---

1: **Input:** Multivariate Time Series $\mathbf{X} \in \mathbb{R}^{B \times C \times T}$, Labels $\mathbf{Y}$
2: **Hyperparameters:** Kernels $\mathcal{K}$, Expansion factor $E$, Regularization weights $\lambda_{SP}, \lambda_{DAG}$
3: **Initialize:** MCE module $\Phi_{MCE}$, TDSSE blocks, SGM modules, Predictor $\Phi_{Pred}$
4: *// 1. Multi-scale Convolutional Embedding (MCE)*
5: $\mathbf{H}_{list} \leftarrow [\ \ ]$
6: **for** $k \in \mathcal{K}$ **do**
7:     $\mathbf{H}_k \leftarrow \text{Conv1D}_k(\mathbf{X})$
8:     $\mathbf{H}_{list}.\text{append}(\mathbf{H}_k)$
9: **end for**
10: $\mathbf{H} \leftarrow \text{Linear}(\text{Concat}(\mathbf{H}_{list}))$
11: $\mathcal{L}_{reg} \leftarrow 0$
12: *// 2. Deep Feature Extraction Loop*
13: **for** $l = 1$ **to** $L$ **do**
14:     $\mathbf{H}_{in} \leftarrow \text{LayerNorm}(\mathbf{H})$
15:     *// (a) Tri-branch Differential Encoder*
16:     $\mathbf{Z}_{raw} \leftarrow \text{SSM}(\mathbf{H}_{in})$
17:     $\mathbf{Z}_{diff} \leftarrow \text{SSM}(\mathbf{H}_{in}[t] - \mathbf{H}_{in}[t-1])$
18:     $\mathbf{Z}_{freq} \leftarrow \text{iFFT}(\text{SpectralGating}(\text{FFT}(\mathbf{H}_{in})))$
19:     $\mathbf{H}_{temp} \leftarrow \text{GatedFusion}(\mathbf{Z}_{raw}, \mathbf{Z}_{diff}, \mathbf{Z}_{freq}) + \mathbf{H}$
20:     *// (b) Spatial Graph Mamba (SGM)*
21:     $\mathbf{A} \leftarrow \text{LearnAdjacency}(\mathbf{H}_{temp})$
22:     $\mathcal{L}_{reg} \leftarrow \mathcal{L}_{reg} + \lambda_{SP}\|\mathbf{A}\|_1 + \lambda_{DAG}\text{tr}(e^{\mathbf{A} \circ \mathbf{A}})$
23:     $\mathbf{H}_{spatial} \leftarrow \text{GraphMamba}(\mathbf{H}_{temp}, \mathbf{A})$
24:     $\mathbf{H} \leftarrow \mathbf{H}_{temp} + \mathbf{H}_{spatial}$
25: **end for**
26: *// 3. Prediction and Loss Calculation*
27: $\mathbf{Z}_{final} \leftarrow \text{GlobalMeanPool}(\mathbf{H})$
28: $\hat{\mathbf{Y}} \leftarrow \Phi_{Pred}(\mathbf{Z}_{final})$
29: $\mathcal{L}_{cls} \leftarrow \text{CrossEntropy}(\hat{\mathbf{Y}}, \mathbf{Y})$
30: $\mathcal{L}_{total} \leftarrow \mathcal{L}_{cls} + \mathcal{L}_{reg}$
31: **Output:** Optimized parameters minimizing $\mathcal{L}_{total}$

---

To comprehensively evaluate the performance of MedMamba and baseline models, we employ six standard evaluation metrics: **Accuracy**, **Precision**, **Recall**, **F1 Score**, **AUROC**, and **AUPRC**. Given that medical datasets often involve multi-class classification and potential class imbalance, we report the **Macro-averaged** scores for all metrics to treat all classes equally, regardless of their support size. The detailed formulations of the metrics are as follows:

- **Accuracy**: Measures the overall proportion of correct predictions across all classes.

$$\text{Accuracy} = \frac{\text{Number of correct predictions}}{\text{Total number of predictions}} \tag{23}$$

where $N$ is the total number of samples.

- **Precision**: It measures the proportion of positive identifications that were actually correct. The macro-averaged precision is:

$$\text{Precision} = \frac{\text{True Positives}}{\text{True Positives} + \text{False Positives}} \tag{24}$$

- **Recall**: It measures the proportion of actual positives that were identified correctly. The macro-averaged recall is:

$$\text{Recall} = \frac{\text{True Positives}}{\text{True Positives} + \text{False Negatives}} \tag{25}$$

- **F1 Score**: The harmonic mean of Precision and Recall, providing a single metric that balances both concerns. It is particularly useful for imbalanced datasets.

$$\text{F1 Score} = \frac{2 \times \text{Precision} \times \text{Recall}}{\text{Precision} + \text{Recall}} \tag{26}$$

- **AUROC (Area Under ROC Curve)**: Assesses the classifier's ability to distinguish between classes across various threshold settings. For multi-class tasks, we adopt a **One-vs-Rest (OvR)** strategy, calculating the area under the ROC curve (plotting TPR vs. FPR) for each class against all others, and then reporting the macro-average.

- **AUPRC (Area Under Precision-Recall Curve)**: Summarizes the Precision-Recall curve, focusing on the trade-off between precision and recall for different thresholds. Similar to AUROC, we calculate the AUPRC for each class in a One-vs-Rest manner and report the macro-average. This metric is often considered more informative than AUROC for highly imbalanced datasets.

## B. Detailed Explanation of Adaptive Graph Learning

In this section, we provide a rigorous derivation and interpretation of the adaptive graph learning mechanism employed in the Spatial Graph Mamba (SGM) module. Specifically, we elaborate on the sample-conditioned generation of the dependency structure and the mathematical justification behind the acyclicity constraint.

### B.1. Sample-Conditioned Graph Generation Mechanism

The core challenge in multivariate medical time series classification is that the functional topology between sensors (e.g., brain regions in EEG or leads in ECG) is often *latent*, *directed*, and highly *dynamic*. Unlike static anatomical graphs which assume fixed relationships, functional dependencies vary significantly across different subjects and even distinct temporal segments within the same recording.

To capture these transient dependencies without relying on prior domain knowledge, MedMamba parameterizes the adjacency matrix $\mathbf{A} \in \mathbb{R}^{C \times C}$ as a dynamic function of the input sample itself, rather than optimizing a static global parameter.

**Feature Abstraction.** Let $\mathbf{Z}_{temp} \in \mathbb{R}^{T \times C \times D}$ denote the temporal features output by the TDSSE module. We first abstract the global characteristics of each channel $c$ by pooling along the temporal dimension, obtaining a concise node representation matrix $\mathbf{U} \in \mathbb{R}^{C \times D}$:

$$\mathbf{u}_c = \text{MeanPool}_t\big(\mathbf{Z}_{temp}[:, c, :]\big) \in \mathbb{R}^D, \tag{27}$$

$$\mathbf{U} = [\mathbf{u}_1; \ldots; \mathbf{u}_C] \in \mathbb{R}^{C \times D}. \tag{28}$$

where the $c$-th row $\mathbf{u}_c$ encapsulates the holistic state of channel $c$ for the current sample.

**Asymmetric Projections for Directedness.** A key requirement for medical signal modeling is capturing **directed** information flow (e.g., a seizure propagating from a focus region to neighbors). A simple symmetric similarity (e.g., cosine similarity of $\mathbf{U}$) implies $\mathbf{A}_{ij} = \mathbf{A}_{ji}$, which fails to model causal-like structures.

To enforce directionality, we project the node representations $\mathbf{U}$ into two distinct latent subspaces: a "Source" space and a "Target" space. Formally, we employ two learnable non-linear projections $\phi_1$ and $\phi_2$:

$$\mathbf{V}_1 = \phi_1(\mathbf{U}) \in \mathbb{R}^{C \times d_{node}}, \quad \mathbf{V}_2 = \phi_2(\mathbf{U}) \in \mathbb{R}^{C \times d_{node}} \tag{29}$$

Here, $\mathbf{V}_2$ represents the capability of each node to *emit* information (source/sender), while $\mathbf{V}_1$ represents the capability to *receive* information (target/receiver). Since $\phi_1 \neq \phi_2$, the resulting interaction is asymmetric, naturally accommodating directed graphs.

**Adaptive Adjacency Generation.** The dependency strength from node $j$ to node $i$ is quantified by the compatibility between the sender embedding of node $j$ in $\mathbf{V}_2$ and the receiver embedding of node $i$ in $\mathbf{V}_1$. We compute the adjacency matrix via a bilinear operation followed by a Sigmoid activation to normalize weights to the range $(0, 1)$:

$$\mathbf{A}_{raw} = \sigma(\mathbf{V}_1 \mathbf{V}_2^\top) \tag{30}$$

where $\sigma(\cdot)$ is the sigmoid function. To eliminate trivial self-correlations which may dominate the graph diffusion process (and to satisfy the strict definition of DAGs which disallow self-loops), we explicitly mask the diagonal elements:

$$\mathbf{A} = \mathbf{A}_{raw} \odot (\mathbf{1} - \mathbf{I}) \tag{31}$$

where $\mathbf{I}$ is the identity matrix.

This formulation ensures that (1) the graph is **adaptive**, as $\mathbf{A}$ changes dynamically with the input $\mathbf{Z}_{temp}$; and (2) the graph is **directed**, as generally $(\mathbf{V}_1 \mathbf{V}_2^\top)_{ij} \neq (\mathbf{V}_1 \mathbf{V}_2^\top)_{ji}$. The resulting $\mathbf{A}$ serves as the pre-normalized adjacency matrix for the subsequent graph diffusion and structure regularization steps.

**Random-walk Normalization for Diffusion.** To perform message diffusion, we construct a row-stochastic transition matrix via random-walk normalization:

$$\mathbf{d} = \mathbf{A}\mathbf{1} \in \mathbb{R}^C, \qquad \widetilde{\mathbf{A}}_{ij} = \frac{\mathbf{A}_{ij}}{\mathbf{d}_i + \varepsilon}, \tag{32}$$

where $\varepsilon > 0$ avoids division by zero. As emphasized in the main text, the structure priors are imposed on the *pre-normalized* adjacency $\mathbf{A}$ rather than $\widetilde{\mathbf{A}}$.

## B.2. Theoretical Justification of Acyclicity Regularization

A key innovation in MedMamba is the imposition of a Directed Acyclic Graph (DAG) constraint. In medical signal propagation, cyclic dependencies (loops) can lead to redundant information feedback and unstable feature amplification during graph propagation. We adopt the trace exponential characterization to enforce acyclicity.

**Theorem (Trace Exponential Condition).** A non-negative adjacency matrix $\mathbf{A} \in \mathbb{R}^{C \times C}$ represents a DAG if and only if:

$$h(\mathbf{A}) = \text{tr}(e^{\mathbf{A}}) - C = 0 \tag{33}$$

**Proof and Derivation.** The matrix exponential $e^{\mathbf{A}}$ can be expanded via its Taylor series:

$$e^{\mathbf{A}} = \sum_{k=0}^{\infty} \frac{\mathbf{A}^k}{k!} = \mathbf{I} + \mathbf{A} + \frac{\mathbf{A}^2}{2!} + \frac{\mathbf{A}^3}{3!} + \dots \tag{34}$$

In graph theory, the $(i, i)$-th entry of the $k$-th power of an adjacency matrix, $(\mathbf{A}^k)_{ii}$, counts the number of paths of length $k$ starting and ending at node $i$ (i.e., cycles of length $k$).

Considering the trace operator $\text{tr}(\cdot)$, which sums the diagonal elements:

$$\text{tr}(e^{\mathbf{A}}) = \text{tr}(\mathbf{I}) + \text{tr}(\mathbf{A}) + \frac{1}{2!}\text{tr}(\mathbf{A}^2) + \sum_{k=3}^{\infty} \frac{1}{k!}\text{tr}(\mathbf{A}^k) \tag{35}$$

Since $\text{tr}(\mathbf{I}) = C$ (where $C$ is the number of channels), the condition $h(\mathbf{A}) = 0$ implies:

$$\sum_{k=1}^{\infty} \frac{1}{k!}\text{tr}(\mathbf{A}^k) = 0 \tag{36}$$

Since $\mathbf{A}$ is non-negative (guaranteed by the Sigmoid activation), all path counts $(\mathbf{A}^k)_{ii}$ are non-negative. Therefore, the sum is zero if and only if $\text{tr}(\mathbf{A}^k) = 0$ for all $k \geq 1$. This implies there are no cycles of any length $k$ in the graph.

In our implementation, to improve numerical stability and gradient flow for weighted graphs, we utilize the Hadamard product form in the regularization term:

$$\mathcal{L}_{dag} = \text{tr}(e^{\mathbf{A} \circ \mathbf{A}}) - C \tag{37}$$

This effectively penalizes the magnitude of the weights along any cyclic path, pushing the structure towards a strict DAG.

### B.3. Optimization and Gradient Flow

The total loss enables end-to-end joint optimization of the classifier and the sample-conditioned graph structure:

$$\mathcal{L}_{total} = \mathcal{L}_{cls} + \lambda_{SP}\|\mathbf{A}\|_1 + \lambda_{DAG}(\text{tr}(e^{\mathbf{A} \circ \mathbf{A}}) - C) \tag{38}$$

- **Sparsity ($\|\mathbf{A}\|_1$):** Medical signals typically exhibit sparse connectivity. The $L_1$ penalty acts on the Sigmoid-activated weights, suppressing weak connections and filtering noise.

- **Gradient Flow:** During backpropagation, the gradients flow from the classification loss through the graph diffusion process back to the projections $\phi_1$ and $\phi_2$. This guides the model to produce sample-specific $\mathbf{A}$ matrices that highlight the most discriminative channel interactions for the specific input instance, subject to the structural constraints imposed by $\mathcal{L}_{dag}$ and $\mathcal{L}_{sp}$.

## C. MedMamba Structure Design and Theoretical Analysis

In this section, we provide the theoretical underpinnings of the MedMamba architecture, focusing on the signal processing properties of the Multi-scale Convolutional Embedding (MCE) and the stationarity-preserving mechanism of the Tri-branch Differential State Space Encoder (TDSSE).

### C.1. Multi-scale Convolutional Embedding as Filter Banks

The raw medical time series $\mathbf{X} \in \mathbb{R}^{C \times T}$ often contains pathological patterns appearing at diverse temporal granularities—for instance, high-frequency spikes in EEG versus low-frequency ST-segment elevation in ECG. A standard single-kernel convolution or linear projection imposes a fixed receptive field, potentially missing scale-variant features.

We formulate the MCE module as a learnable **filter bank**. Let $\mathbf{x} \in \mathbb{R}^T$ be a univariate input channel. The operation with a kernel size $k$ can be viewed as a Finite Impulse Response (FIR) filter:

$$\mathbf{h}^{(k)}[t] = (\mathbf{x} * \mathbf{w}^{(k)})[t] = \sum_{\tau=0}^{k-1} \mathbf{w}^{(k)}[\tau]\mathbf{x}[t - \tau] \tag{39}$$

By employing a set of kernels $\mathcal{K} = \{k_1, k_2, \ldots, k_m\}$ with varying receptive fields, MedMamba performs a **Multi-Resolution Analysis (MRA)** analogous to a discrete wavelet transform.

- Small kernels (e.g., $k = 3$) act as high-pass filters, capturing transient local variations (noise or sharp onsets).

- Large kernels (e.g., $k = 7$) act as smoother low-pass filters, capturing broader morphological trends.

The concatenation $\mathbf{H} = \text{Concat}(\{\mathbf{h}^{(k)}\}_{k \in \mathcal{K}})$ ensures that the subsequent SSM backbone receives a representation rich in both transient and trend information.

## C.2. Theoretical Basis of the Tri-branch Design

Standard State Space Models (SSMs) like Mamba are defined by the linear recurrence $\mathbf{h}_t = \mathbf{A}\mathbf{h}_{t-1} + \mathbf{B}\mathbf{x}_t$. While efficient, this formulation assumes that the input statistics are relatively stationary over time. Medical data, however, suffers from **non-stationarity** (distribution shifts), such as baseline drift caused by patient movement or sensor impedance changes.

The TDSSE addresses this via three complementary views. We analyze their theoretical roles below:

**1. The Differential View: Stationarity via Differencing.** Baseline drift can be modeled as an additive trend component $\mathbf{T}_t$:

$$\mathbf{x}_t = \mathbf{s}_t + \mathbf{T}_t + \epsilon_t \tag{40}$$

where $\mathbf{s}_t$ is the true physiological signal and $\mathbf{T}_t$ is a slowly varying trend (e.g., a low-frequency drift). A standard SSM might overfit to $\mathbf{T}_t$. The differential branch inputs the first-order difference $\Delta\mathbf{x}_t = \mathbf{x}_t - \mathbf{x}_{t-1}$.

$$\Delta\mathbf{x}_t = (\mathbf{s}_t - \mathbf{s}_{t-1}) + (\mathbf{T}_t - \mathbf{T}_{t-1}) + \Delta\epsilon_t \tag{41}$$

Since the trend $\mathbf{T}_t$ is slowly varying, $\mathbf{T}_t \approx \mathbf{T}_{t-1}$, implying $\Delta\mathbf{T}_t \approx 0$. Thus, $\Delta\mathbf{x}_t \approx \Delta\mathbf{s}_t + \Delta\epsilon_t$.

**Theoretical Implication:** The differential branch effectively acts as a high-pass filter that suppresses non-stationary trends, providing a stable input distribution for the SSM to model intrinsic dynamics.

**2. The Frequency View: Global Spectral Correlations.** While SSMs capture long-range dependencies recurrently ($O(L)$), they can still struggle with global periodicities due to the "vanishing memory" of the recurrent state $\mathbf{h}_t$. The frequency branch leverages the Fourier Transform's global property. By computing $\mathcal{F}(\mathbf{x})$, every point in the frequency domain depends on all time steps $t = 1 \ldots T$:

$$\mathbf{X}[k] = \sum_{n=0}^{T-1} \mathbf{x}[n] e^{-i2\pi kn/T} \tag{42}$$

We apply a learnable spectral gate $\mathbf{G} \in \mathbb{C}^T$ via element-wise multiplication $\mathbf{Y}[k] = \mathbf{X}[k] \odot \mathbf{G}[k]$, which corresponds to a circular convolution in the time domain with a filter of size $T$.

**Theoretical Implication:** This branch provides a "shortcut" for global context, allowing the model to explicitly amplify or suppress specific physiological frequency bands (e.g., Alpha waves in EEG) irrespective of their temporal position.

## C.3. Gated Fusion Mechanism

To integrate these divergent views, we employ a Gated Fusion mechanism. Let $\mathbf{Z}_{raw}, \mathbf{Z}_{diff}, \mathbf{Z}_{freq}$ be the representations from the three branches. The fused output $\mathbf{H}_{out}$ is derived as:

$$\mathbf{H}_{out} = \mathbf{Z}_{raw} \odot \sigma(\mathbf{W}_g\mathbf{Z}_{combined}) + \mathbf{Z}_{diff} \odot (1 - \sigma(\mathbf{W}_g\mathbf{Z}_{combined})) + \mathbf{Z}_{freq} \tag{43}$$

where $\sigma$ is the Sigmoid function. Mathematically, this acts as a **soft-switching mechanism**:

- In regions with severe baseline drift, the gate can prioritize $\mathbf{Z}_{diff}$ (stationarity) over $\mathbf{Z}_{raw}$.

- In regions requiring precise waveform reconstruction, it can weigh $\mathbf{Z}_{raw}$ higher.

This allows MedMamba to dynamically adapt its structural bias based on the local signal characteristics.

## C.4. Addition: Bidirectional Selective State Space Modeling (Bi-Mamba)

Let $\mathbf{H} \in \mathbb{R}^{T \times C \times D}$ denote the input sequence features, where $\mathbf{h}_{t,c} \in \mathbb{R}^D$ is the feature at time step $t$ of channel $c$. Bi-Mamba augments a selective state space layer with both forward (left-to-right) and backward (right-to-left) scans, so that each output token can leverage context from both past and future.

**Continuous-time SSM and discretization.** We start from the standard continuous-time linear state space model

$$\frac{d\mathbf{x}(t)}{dt} = \mathbf{A}\mathbf{x}(t) + \mathbf{B}\mathbf{u}(t), \qquad \mathbf{y}(t) = \mathbf{C}\mathbf{x}(t) + \mathbf{D}\mathbf{u}(t), \tag{44}$$

where $\mathbf{x}(t) \in \mathbb{R}^N$ is the latent state, $\mathbf{u}(t) \in \mathbb{R}^D$ is the input, and $\mathbf{y}(t) \in \mathbb{R}^D$ is the output. Applying a zero-order hold (ZOH) discretization with step size $\Delta_{t,c} > 0$ yields the per-step recurrence

$$\mathbf{x}_{t,c} = \overline{\mathbf{A}}_{t,c}\mathbf{x}_{t-1,c} + \overline{\mathbf{B}}_{t,c}\mathbf{u}_{t,c}, \qquad \mathbf{y}_{t,c} = \mathbf{C}\mathbf{x}_{t,c} + \mathbf{D}\mathbf{u}_{t,c}, \tag{45}$$

with

$$\overline{\mathbf{A}}_{t,c} = \exp(\Delta_{t,c}\mathbf{A}), \qquad \overline{\mathbf{B}}_{t,c} = (\exp(\Delta_{t,c}\mathbf{A}) - \mathbf{I})\,\mathbf{A}^{-1}\mathbf{B}. \tag{46}$$

**Selective (input-conditioned) parameterization.** Following the selective SSM design, we condition the discretization step size (and optionally gates) on the current token:

$$\mathbf{u}_{t,c} = \mathbf{W}_u\mathbf{h}_{t,c}, \qquad \Delta_{t,c} = \mathrm{softplus}\big(\mathbf{w}_\Delta^\top\mathbf{h}_{t,c} + b_\Delta\big), \tag{47}$$

and employ a multiplicative output gate

$$\mathbf{g}_{t,c} = \sigma(\mathbf{W}_g\mathbf{h}_{t,c}), \qquad \mathbf{o}_{t,c} = \mathbf{g}_{t,c} \odot \mathbf{y}_{t,c}, \tag{48}$$

where $\sigma(\cdot)$ is the sigmoid function and $\odot$ denotes element-wise multiplication. This conditioning enables the effective memory timescale to adapt to content, which is beneficial for nonstationary physiological dynamics.

**Bidirectional scanning.** To incorporate both causal and anti-causal context, we compute two selective scans.

*Forward scan (left-to-right).*

$$\mathbf{x}_{t,c}^\rightarrow = \overline{\mathbf{A}}_{t,c}\mathbf{x}_{t-1,c}^\rightarrow + \overline{\mathbf{B}}_{t,c}\mathbf{u}_{t,c}, \qquad \mathbf{o}_{t,c}^\rightarrow = \mathbf{g}_{t,c} \odot \big(\mathbf{C}\mathbf{x}_{t,c}^\rightarrow + \mathbf{D}\mathbf{u}_{t,c}\big). \tag{49}$$

*Backward scan (right-to-left).* Define the reversed sequence $\widetilde{\mathbf{h}}_{t,c} = \mathbf{h}_{T-t+1,c}$ and compute

$$\mathbf{x}_{t,c}^\leftarrow = \widetilde{\overline{\mathbf{A}}}_{t,c}\mathbf{x}_{t-1,c}^\leftarrow + \widetilde{\overline{\mathbf{B}}}_{t,c}\widetilde{\mathbf{u}}_{t,c}, \qquad \widetilde{\mathbf{o}}_{t,c}^\leftarrow = \widetilde{\mathbf{g}}_{t,c} \odot \big(\mathbf{C}\mathbf{x}_{t,c}^\leftarrow + \mathbf{D}\widetilde{\mathbf{u}}_{t,c}\big), \tag{50}$$

where $\widetilde{\mathbf{u}}_{t,c} = \mathbf{W}_u\widetilde{\mathbf{h}}_{t,c}$ and $\widetilde{\Delta}_{t,c} = \mathrm{softplus}(\mathbf{w}_\Delta^\top\widetilde{\mathbf{h}}_{t,c} + b_\Delta)$ are defined analogously, and thus $\widetilde{\overline{\mathbf{A}}}_{t,c}, \widetilde{\overline{\mathbf{B}}}_{t,c}$ follow the same discretization as above. We then restore the original temporal order by

$$\mathbf{o}_{t,c}^\leftarrow = \widetilde{\mathbf{o}}_{T-t+1,c}^\leftarrow. \tag{51}$$

**Fusion.** Finally, we fuse the forward and backward representations to obtain the bidirectional output:

$$\mathbf{z}_{t,c} = \mathbf{W}_o\big[\mathbf{o}_{t,c}^\rightarrow; \mathbf{o}_{t,c}^\leftarrow\big] + \mathbf{b}_o \in \mathbb{R}^D, \tag{52}$$

where $[\cdot; \cdot]$ denotes concatenation.[3]

# D. Computational Complexity Analysis

In this section, we provide a formal complexity analysis of MedMamba compared to standard architectures like Transformers and CNNs. We analyze the computational costs with respect to the sequence length $T$, the number of channels $C$, and the hidden dimension $D$.

## D.1. Overall Complexity Comparison

Table 4 summarizes the theoretical time complexity of MedMamba against mainstream baselines. The key advantage of MedMamba is its linear complexity with respect to the sequence length $T$, making it suitable for long-duration medical monitoring.

---

[3]A simpler alternative is $\mathbf{z}_{t,c} = \mathbf{o}_{t,c}^\rightarrow + \mathbf{o}_{t,c}^\leftarrow$; we adopt concatenation for greater expressivity.

*Table 4.* Computational complexity comparison. $T$: Sequence length, $C$: Number of channels, $D$: Hidden dimension, $K$: Kernel size, $N$: SSM state dimension.

| Model | Time Complexity | Space Complexity |
|---|:---:|:---:|
| Standard Transformer | $\mathcal{O}(C \cdot T^2 \cdot D + C^2 \cdot T \cdot D)$ | $\mathcal{O}(T^2 + CD)$ |
| Standard CNN (1D) | $\mathcal{O}(C \cdot T \cdot K \cdot D^2)$ | $\mathcal{O}(T \cdot CD)$ |
| **MedMamba (Ours)** | $\mathcal{O}(C \cdot T \log T \cdot D + T \cdot C^2 \cdot D)$ | $\mathcal{O}(T \cdot CD + C^2)$ |

### D.2. Detailed Breakdown by Module

**1. Multi-scale Convolutional Embedding (MCE).** The MCE module consists of parallel 1D depthwise convolutions. For a set of kernels with maximum size $K$, the complexity is:

$$\mathcal{O}_{MCE} = \mathcal{O}(T \cdot C \cdot D \cdot K) \tag{53}$$

Since $K$ is a small constant (e.g., $K = 7$) and independent of $T$, this operation is strictly linear $\mathcal{O}(T)$.

**2. Tri-branch Differential State Space Encoder (TDSSE).** This module has three parallel branches:

- **Raw & Differential Branches (SSM):** The core Mamba (SSM) operation relies on a parallel scan algorithm. With state dimension $N$, the complexity is $\mathcal{O}(T \cdot D \cdot N)$. Since $N$ is constant (typically 16), this is linear $\mathcal{O}(T)$.

- **Frequency Branch (FFT):** The Fast Fourier Transform (FFT) and inverse FFT require $\mathcal{O}(T \log T)$ operations per channel. Thus, this branch contributes $\mathcal{O}(C \cdot D \cdot T \log T)$.

Total TDSSE Complexity:

$$\mathcal{O}_{TDSSE} \approx \mathcal{O}(C \cdot D \cdot T \log T) \tag{54}$$

While strictly *quasi-linear* due to the $\log T$ term, in practice, for physiological signal lengths, $T \log T$ is comparable to linear scaling and significantly faster than the quadratic $\mathcal{O}(T^2)$ of Attention.

**3. Spatial Graph Mamba (SGM).** The SGM module handles channel interactions:

- **Graph Learning:** Computing the adjacency matrix $\mathbf{A} \in \mathbb{R}^{C \times C}$ from node embeddings takes $\mathcal{O}(C^2 \cdot d_{node})$. The acyclicity regularization (matrix exponential) takes $\mathcal{O}(C^3)$ but is computed only once per batch/step, not per time-point $t$.

- **Graph Propagation:** Assuming a dense adjacency matrix learned from data, the spatial aggregation at each time step involves multiplying $\mathbf{A}$ (size $C \times C$) with features $\mathbf{H}_t$ (size $C \times D$). This yields $\mathcal{O}(T \cdot C^2 \cdot D)$.

Total SGM Complexity:

$$\mathcal{O}_{SGM} = \mathcal{O}(T \cdot C^2 \cdot D + C^3) \tag{55}$$

In medical settings, the channel count $C$ is typically small, whereas $T$ is large. Thus, the quadratic term in $C$ is computationally negligible compared to the reduction in $T$.

To sum up, combining these components, the dominant term for MedMamba is $\mathcal{O}(T \log T)$ due to the FFT branch, or $\mathcal{O}(T)$ if the FFT branch is disabled. This confirms that MedMamba scales effectively linearly with sequence length, addressing the bottleneck of Transformer-based baselines on high-sampling-rate medical recordings.

*Table 5.* **Classification results of our MedMamba and baseline models on the ADFTD dataset under the subject-independent setting**. The best results are highlighted in **red**, and the second-best in **blue**. Note that the percentage symbol is omitted. Since the source code for **KEMed** is not publicly available, we could not reproduce its results on this dataset. The results for **GPT4TS** are based on our reproduction using its official code and others are from the original reports.

| Dataset | Metric | Ours MedMamba | WWW'25 MedGNN | ACM MM'25 KEMed | Neurips'24 MedFormer | ICLR'24 iTransformer | Neurips'24 FourierGNN | InfoS'24 TodyNet | IJCAI'24 MTST | ICLR'23 PatchTST | Neurips'23 GPT4TS |
|---|---|---|---|---|---|---|---|---|---|---|---|
| ADFTD | Accuracy | **98.66**±0.16 | **98.15**±0.25 | - | 95.82±0.53 | 90.71±0.47 | 83.33±1.13 | 97.23±0.35 | 83.75±0.58 | 93.98±0.40 | 97.87±0.19 |
| | Precision | **98.66**±0.17 | **98.16**±0.23 | - | 95.87±0.51 | 90.76±0.43 | 83.56±1.16 | 97.26±0.34 | 84.14±0.54 | 94.03±0.40 | 97.87±0.19 |
| | Recall | **98.68**±0.15 | **98.18**±0.26 | - | 95.84±0.54 | 90.72±0.44 | 83.34±1.17 | 97.24±0.35 | 83.76±0.58 | 93.99±0.40 | 97.89±0.19 |
| | F1 Score | **98.67**±0.16 | **98.16**±0.25 | - | 95.84±0.52 | 90.72±0.44 | 83.21±1.18 | 97.24±0.35 | 83.67±0.57 | 93.99±0.40 | 97.87±0.19 |
| | AUROC | **99.88**±0.02 | **99.80**±0.05 | - | 99.39±0.13 | 97.43±0.18 | 93.30±0.57 | 99.72±0.04 | 94.61±0.23 | 98.71±0.10 | 99.79±0.03 |
| | AUPRC | **99.86**±0.03 | **99.78**±0.05 | - | 99.25±0.19 | 96.53±0.33 | 89.26±0.94 | 99.64±0.07 | 90.96±0.38 | 98.39±0.15 | 99.75±0.03 |

# E. More Experiment Results

## E.1. Subject-Dependent Evaluation

## E.2. Subject-Independent Evaluation Analysis

In this study, we rigorously adopt a **subject-independent split strategy** for model training and evaluation. Under this protocol, the training, validation, and test sets are strictly partitioned by subject ID, ensuring that samples from the same individual appear exclusively in one subset. This setting closely mirrors real-world clinical diagnostic scenarios, where models trained on a cohort of historical patients must generalize to previously unseen individuals, thereby testing the model's ability to learn robust, subject-invariant pathological features rather than overfitting to individual-specific biometrics.

Table 2 presents a comprehensive performance comparison between our proposed MedMamba and ten representative state-of-the-art models across real-world physiological datasets. Overall, MedMamba demonstrates dominant performance, achieving the **best results across all metrics** on almost all datasets. Notably, on the challenging ADFTD dataset, MedMamba surpasses the second-best method (MedGNN) by significant margins in Accuracy and AUROC, proving its superior modeling capability and robustness in complex multi-class scenarios. We further analyze the performance gaps by categorizing the baselines into three distinct families:

**Comparison with Graph-based Methods.** Models like **MedGNN** and **TodyNet** explicitly model inter-channel relationships, with MedGNN achieving strong second-place performance in our experiments. While these methods effectively capture spatial correlations, they often rely on heavy dynamic graph computations or predefined adjacency structures. In contrast, MedMamba introduces a lightweight *Spatial Graph Mamba* module that learns a directed acyclic graph (DAG) structure. This not only uncovers latent causal dependencies between sensors but also imposes sparsity constraints, filtering out noise. Furthermore, standard GNNs often lack specialized mechanisms for non-stationary handling, whereas MedMamba's tri-branch design explicitly mitigates baseline drift, giving it a decisive edge in handling real-world medical signals.

**Comparison with Large Model-based Methods.** **GPT4TS** represents the emerging trend of leveraging pre-trained Large Language Models (LLMs) for time series analysis. While it achieves competitive results by exploiting the universality of pre-trained frozen transformers, it suffers from two limitations: (1) high computational cost due to the massive parameter size, and (2) a lack of domain-specific inductive biases for multi-channel biological signals. MedMamba, conversely, achieves superior accuracy with a fraction of the computational budget. By integrating domain-specific designs—such as the differential branch for stationarity and multi-scale embeddings for morphology—MedMamba proves that a specialized, lightweight architecture can outperform general-purpose heavy models in medical domains.

**Comparison with Transformer and Frequency-based Methods.** Transformer variants like **iTransformer**, **MedFormer**, and **PatchTST** excel at capturing long-range dependencies via self-attention. However, they face a quadratic complexity bottleneck regarding sequence length and often struggle to balance local morphological details with global context. Frequency-based methods like **FourierGNN** attempt to address global dependencies via spectral projections but may lose transient time-domain information crucial for detecting anomalies like spikes. MedMamba bridges these gaps by combining the linear efficiency of State Space Models with a *Tri-branch Differential Encoder*. This allows it to simultaneously model local morphology (via Convolutional Embeddings), global periodicity (via the Frequency branch), and long-term dependencies (via the SSM backbone), resulting in a more holistic and robust representation.

## E.3. Component Ablation Results

*Table 6.* **Comprehensive component ablation analysis across APAVA, PTB-XL, and TDBRAIN datasets.** We report mean±std. ↓ △ indicates the performance drop relative to the full MedMamba. Column headers "**w/o MCE**", "**w/o TDSSE**", and "**w/o SGM**" denote the removal of the Multi-scale Convolutional Embedding, Tri-branch Differential Encoder, and Spatial Graph Mamba module, respectively.

| Metric | APAVA | | | | PTB-XL | | | | TDBRAIN | | | |
|---|---|---|---|---|---|---|---|---|---|---|---|---|
| | Full | w/o MCE | w/o TDSSE | w/o SGM | Full | w/o MCE | w/o TDSSE | w/o SGM | Full | w/o MCE | w/o TDSSE | w/o SGM |
| Accuracy | 84.48±0.55 | 82.97±0.64 (↓1.51) | 80.41±0.79 (↓4.07) | 81.58±0.71 (↓2.90) | 73.86±0.30 | 72.93±0.36 (↓0.93) | 70.46±0.47 (↓3.40) | 71.81±0.40 (↓2.05) | 92.13±0.41 | 91.07±0.53 (↓1.06) | 89.12±0.79 (↓3.01) | 90.62±0.64 (↓1.51) |
| Precision | 86.79±0.62 | 85.34±0.73 (↓1.45) | 82.69±0.86 (↓4.10) | 84.02±0.80 (↓2.77) | 66.15±0.67 | 65.04±0.76 (↓1.11) | 61.62±0.96 (↓4.53) | 63.91±0.81 (↓2.24) | 92.33±0.37 | 91.24±0.50 (↓1.09) | 89.31±0.74 (↓3.02) | 90.79±0.60 (↓1.54) |
| Recall | 81.87±0.27 | 80.08±0.38 (↓1.79) | 76.83±0.57 (↓5.04) | 78.37±0.50 (↓3.50) | 61.30±1.08 | 60.33±1.16 (↓0.97) | 56.71±1.34 (↓4.59) | 58.94±1.20 (↓2.36) | 92.13±0.41 | 91.06±0.54 (↓1.07) | 89.08±0.82 (↓3.05) | 90.57±0.66 (↓1.56) |
| F1 Score | 83.03±0.49 | 81.31±0.60 (↓1.72) | 78.86±0.74 (↓4.17) | 80.05±0.66 (↓2.98) | 63.20±0.91 | 62.26±0.97 (↓0.94) | 58.94±1.13 (↓4.26) | 61.11±1.00 (↓2.09) | 92.12±0.42 | 91.04±0.52 (↓1.08) | 89.05±0.78 (↓3.07) | 90.55±0.63 (↓1.57) |
| AUROC | 86.30±0.17 | 85.19±0.24 (↓1.11) | 83.08±0.36 (↓3.22) | 84.01±0.31 (↓2.29) | 89.33±0.22 | 88.71±0.29 (↓0.62) | 87.92±0.31 (↓1.41) | 88.10±0.33 (↓1.23) | 95.62±0.66 | 94.63±0.76 (↓0.99) | 92.86±0.98 (↓2.76) | 93.84±0.87 (↓1.78) |
| AUPRC | 87.07±0.99 | 85.68±1.08 (↓1.39) | 82.79±1.33 (↓4.28) | 84.29±1.19 (↓2.78) | 66.91±0.75 | 65.84±0.83 (↓1.07) | 62.58±0.99 (↓4.33) | 64.77±0.87 (↓2.14) | 96.68±0.63 | 95.77±0.70 (↓0.91) | 93.81±0.92 (↓2.87) | 94.98±0.81 (↓1.70) |

We provide the component ablation results for the remaining datasets in Table 6. The results show that the full MedMamba consistently achieves the best results across all settings. Removing different modules sequentially leads to varying degrees of performance degradation, demonstrating the effectiveness of our design.

## E.4. MCE Ablation Results

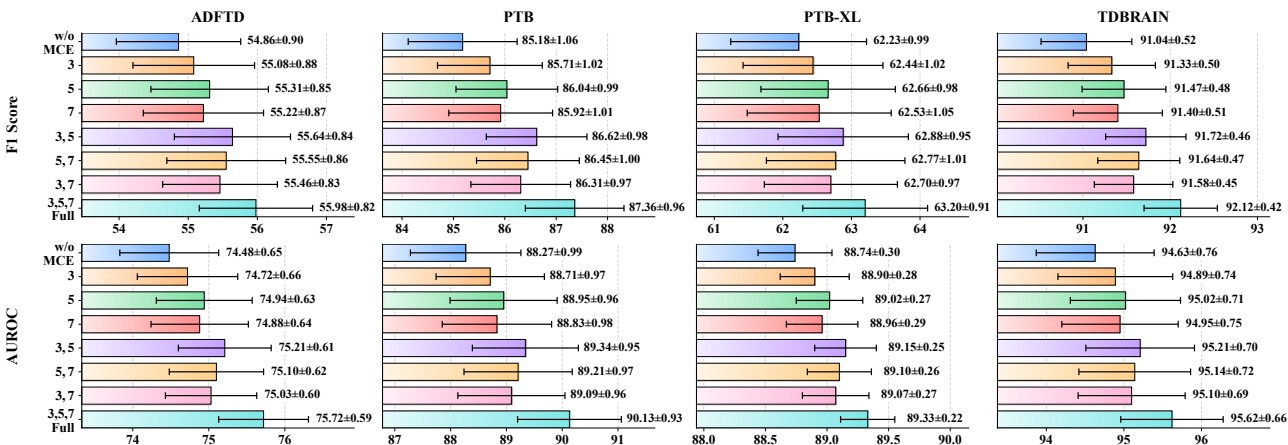

*Figure 10.* **MCE ablation on other datasets under the SI setting.** From left to right, the results are for the ADFTD, PTB, PTB-XL, and TDBRAIN datasets, respectively. The top row shows the F1 Score, and the bottom row shows the AUROC metric. Performance of different convolutional kernel configurations in MCE.

## E.5. Tri-Branch Ablation Results

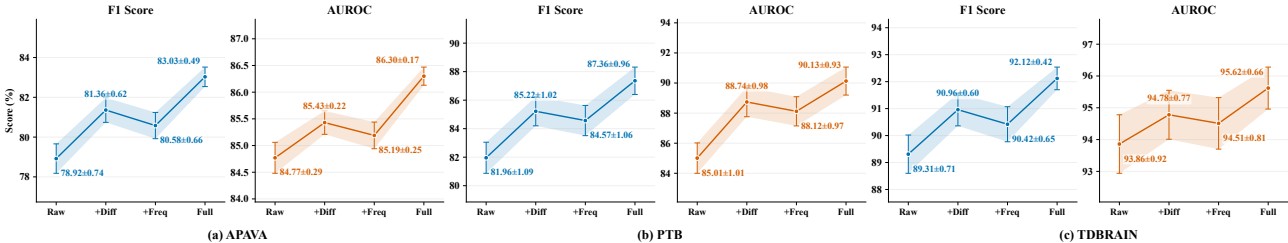

*Figure 11.* **Tri-branch ablation results under the SI setting on (a) APAVA, (b) PTB and (c) TDBRAIN.**

### E.6. SGM Ablation Results

*Table 7.* **SGM ablation on ADFTD, PTB and PTB-XL under the SI setting.** We compare the **Full** model with three variants: **"Fixed Graph"** (no learning), **"w/o $\lambda_{SP}$"** (sparsity prior), and **"w/o $\lambda_{DAG}$"** (DAG prior). We report mean±std. $\downarrow \triangle$ indicates the performance drop relative to the full model.

| Metric | ADFTD | | | | PTB | | | | PTB-XL | | | |
|---|---|---|---|---|---|---|---|---|---|---|---|---|
| | **Full** | **Fixed Graph** | **w/o $\mathcal{L}_{SP}$** | **w/o $\mathcal{L}_{DAG}$** | **Full** | **Fixed Graph** | **w/o $\mathcal{L}_{SP}$** | **w/o $\mathcal{L}_{DAG}$** | **Full** | **Fixed Graph** | **w/o $\mathcal{L}_{SP}$** | **w/o $\mathcal{L}_{DAG}$** |
| **F1 Score** | **55.98**±0.82 | 53.62±0.94 (↓2.36) | 54.41±0.88 (↓1.57) | 54.74±0.86 (↓1.24) | **87.36**±0.96 | 84.92±1.05 (↓2.44) | 86.18±1.00 (↓1.18) | 86.56±0.98 (↓0.80) | **63.20**±0.91 | 61.02±1.03 (↓2.18) | 62.14±0.98 (↓1.06) | 62.46±0.95 (↓0.74) |
| **AUROC** | **75.72**±0.59 | 73.28±0.72 (↓2.44) | 74.10±0.66 (↓1.62) | 74.46±0.63 (↓1.26) | **90.13**±0.93 | 86.71±1.08 (↓3.42) | 88.76±0.99 (↓1.37) | 89.17±0.95 (↓0.96) | **89.33**±0.22 | 88.04±0.35 (↓1.29) | 88.69±0.27 (↓0.64) | 88.92±0.26 (↓0.41) |

### E.7. $\lambda_{SP}$ & $\lambda_{DAG}$ Analysis

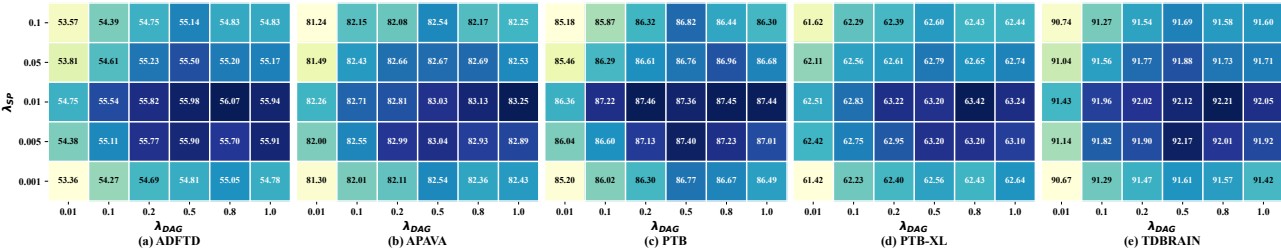

*Figure 12.* **Hyperparameter sensitivity analysis regarding $\lambda_{SP}$ and $\lambda_{DAG}$.** We illustrate the F1 score variations under different combinations of the sparsity regularization $\lambda_{SP}$ and the DAG constraint $\lambda_{DAG}$ across five datasets: (a) ADFTD, (b) APAVA, (c) PTB, (d) PTB-XL, and (e) TDBRAIN. Darker colors indicate higher performance. The results demonstrate consistent robustness, with the optimal performance generally achieved around $\lambda_{SP} = 0.01$ and $\lambda_{DAG} = 0.5$.

To evaluate the sensitivity of the proposed method to the hyperparameters $\lambda_{SP}$ and $\lambda_{DAG}$, we performed a comprehensive grid search on all datasets. The results, depicted as heatmaps, reveal that the model is relatively sensitive to the sparsity penalty $\lambda_{SP}$, consistently achieving optimal performance at $0.01$ (represented by the darkest regions) across varying data distributions; values deviating from this optimum lead to performance drops. Regarding the DAG constraint $\lambda_{DAG}$, the performance improves initially with the increase of the coefficient and reaches a plateau around $0.5$, indicating that the learned graph structure becomes stable beyond this threshold. Consequently, we adopt $\lambda_{SP} = 0.01$ and $\lambda_{DAG} = 0.5$ as the default configuration for all subsequent experiments to ensure both robustness and generalization.

*Table 8.* Performance (mean±std) under the SI setting. **MedMamba** results are from Table 2. **Affirm** and **S-D-Mamba** are Mamba-based baselines adapted for classification.

| Dataset | Model | Accuracy | Precision | Recall | F1 Score | AUROC | AUPRC |
|---|---|---|---|---|---|---|---|
| ADFTD | **MedMamba** (Ours) | **57.56**±0.93 | **56.01**±0.81 | **57.41**±0.79 | **55.98**±0.82 | **75.72**±0.59 | **58.85**±0.67 |
| | Affirm (AAAI'25) | 54.12±0.85 | 53.21±0.76 | 54.00±0.82 | 52.88±0.79 | 72.64±0.63 | 55.37±0.74 |
| | S-D-Mamba (NeuCom'25) | 53.58±0.88 | 52.72±0.80 | 53.31±0.85 | 52.29±0.81 | 71.98±0.67 | 54.72±0.78 |
| APAVA | **MedMamba** (Ours) | **84.48**±0.55 | **86.79**±0.62 | **81.87**±0.27 | **83.03**±0.49 | **86.30**±0.17 | **87.07**±0.99 |
| | Affirm (AAAI'25) | 82.21±0.61 | 84.57±0.70 | 79.62±0.33 | 80.97±0.55 | 84.64±0.21 | 84.96±1.07 |
| | S-D-Mamba (NeuCom'25) | 81.76±0.65 | 84.02±0.75 | 79.05±0.36 | 80.41±0.58 | 84.31±0.24 | 84.41±1.12 |
| PTB | **MedMamba** (Ours) | **89.36**±0.45 | **90.01**±0.48 | **85.70**±1.02 | **87.36**±0.96 | **90.13**±0.93 | **88.73**±1.11 |
| | Affirm (AAAI'25) | 85.71±0.55 | 87.03±0.59 | 81.94±1.08 | 84.07±0.98 | 86.92±0.99 | 85.42±1.18 |
| | S-D-Mamba (NeuCom'25) | 86.14±0.52 | 87.47±0.56 | 82.51±1.06 | 84.53±0.97 | 87.36±0.97 | 85.97±1.14 |
| PTB-XL | **MedMamba** (Ours) | **73.86**±0.30 | **66.15**±0.67 | **61.30**±1.08 | **63.20**±0.91 | **89.33**±0.22 | **66.91**±0.75 |
| | Affirm (AAAI'25) | 71.34±0.38 | 63.92±0.76 | 58.64±1.12 | 59.94±0.96 | 87.45±0.27 | 63.45±0.84 |
| | S-D-Mamba (NeuCom'25) | 70.68±0.41 | 63.15±0.80 | 57.92±1.15 | 59.19±0.99 | 87.11±0.30 | 62.73±0.87 |
| TDBRAIN | **MedMamba** (Ours) | **92.13**±0.41 | **92.33**±0.37 | **92.13**±0.41 | **92.12**±0.42 | **95.62**±0.66 | **96.68**±0.63 |
| | Affirm (AAAI'25) | 90.28±0.47 | 90.49±0.44 | 90.23±0.49 | 90.21±0.46 | 93.69±0.73 | 94.83±0.68 |
| | S-D-Mamba (NeuCom'25) | 89.91±0.50 | 90.12±0.47 | 89.86±0.52 | 89.84±0.49 | 93.21±0.78 | 94.32±0.72 |

### E.8. Comparison with Mamba Baseline

To maintain consistency with previous work (Wang et al., 2024; Fan et al., 2025), we did not present results based on Mamba in the main text. Additionally, considering that the code for similar works like EEGMamba (Gui et al., 2024) and TSCMamba (Ahamed & Cheng, 2025) is not open-source, we used two Mamba-based baselines (Affirm (Wu et al., 2025) and S-D-Mamba (Wang et al., 2025)) for comparison. Table 8 provide the detailed results.

MedMamba consistently outperforms the two Mamba-based baselines because it injects medical time-series–specific inductive biases that are not present in generic Mamba adaptations. Affirm enhances Mamba with frequency filtering and multi-scale interaction, which can improve robustness to periodic components, but it does not explicitly address baseline drift via a differential view nor model sample-specific cross-channel dependencies. S-D-Mamba emphasizes inter-variable correlation modeling with a bidirectional Mamba encoder, yet it is primarily designed for forecasting-style representations and similarly lacks explicit multi-view nonstationarity handling and adaptive spatial dependency learning. In contrast, MedMamba jointly (i) preserves local morphology with multi-scale convolutions, (ii) mitigates nonstationarity through raw/difference/frequency temporal views with gated fusion, and (iii) learns a sample-conditioned dependency structure with sparsity and acyclicity priors, yielding more robust representations for multichannel clinical classification.

### E.9. Qualitative Analysis of Spatial Dependencies

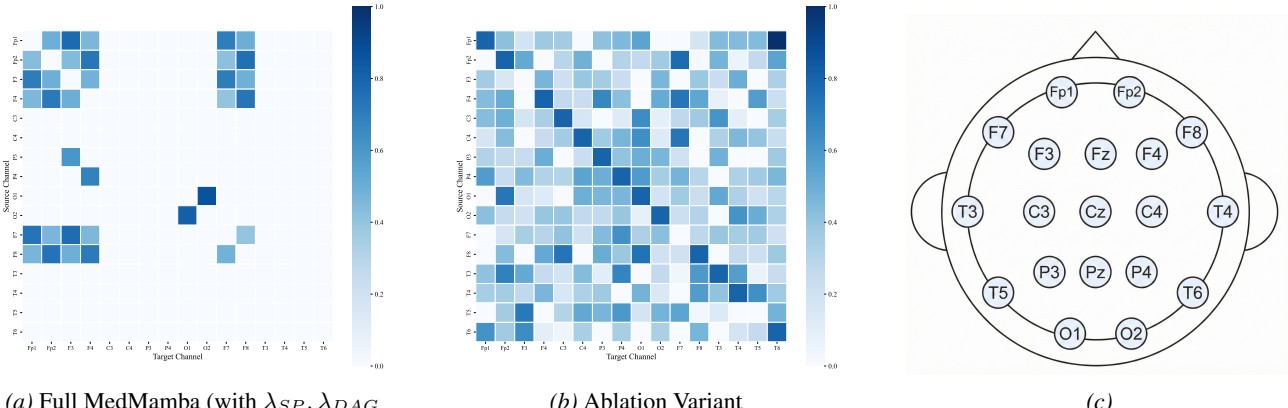

*(a)* Full MedMamba (with $\lambda_{SP}, \lambda_{DAG}$            *(b)* Ablation Variant                                    *(c)*

*Figure 13.* **Visualization of learned adjacency matrices on the APAVA dataset.** We compare the directed dependency structure learned by (a) the full MedMamba model with structure priors ($\lambda_{SP}, \lambda_{DAG}$) and (b) the ablation variant without regularization. The x-axis represents the target channels, and the y-axis represents the source channels. (c) Physiological Placement Diagram of EEG Channels. **(a)** The full model exhibits a clear **sparse topology** with distinct functional clusters, such as the strong connectivity within the Occipital region (O1, O2) and Frontal region (Fp1-F4). **(b)** In contrast, the unregularized variant produces a **dense and noisy** graph (over-smoothing), failing to capture discriminative spatial patterns.

To validate the interpretability of the proposed SGM module, we visualize the learned adjacency matrix $\mathcal{A}$ on the APAVA dataset in Figure 13. By mapping the learned weights to the physiological electrode positions (Standard 10-20 System), we observe that MedMamba captures biologically plausible functional connectivity rather than statistical noise.

**Impact of Structure Priors.**    Comparing Figure 13(a) and (b), the effect of our regularization is evident. The full model learns a **highly sparse topology** where irrelevant connections are suppressed (indicated by white regions), indicating that the sparsity prior ($\lambda_{SP}$) successfully filters out spurious correlations. In contrast, the ablation variant yields a dense, "checkerboard-like" matrix with high entropy. This suggests that without explicit constraints, the graph module suffers from **over-smoothing**, treating all channel interactions as equally important, which dilutes diagnostic information.

**Physiological Interpretability.**    The non-zero entries in the full model (Figure 13(a)) reveal structured functional modules that align with brain anatomy:

- **Local Functional Clustering:** We observe strong interactions between spatially adjacent channels. For instance, the **Occipital region (O1, O2)** exhibits a prominent dense block (bottom-right of the diagonal), reflecting the tight functional coupling of the visual cortex. Similarly, the **Pre-frontal (Fp1, Fp2)** and **Frontal (F3, F4)** channels form

localized clusters, consistent with the volume conduction effect and local neural synchronization.

- **Long-range Connectivity:** Crucially, the model uncovers specific long-range directed pathways, such as the connections from the **Parietal/Temporal regions** projecting to the **Frontal** areas. In the context of Alzheimer's Disease (APAVA dataset), such disruptions or specific patterns in frontoparietal connectivity are often cited as biomarkers of cognitive decline.

In summary, MedMamba does not merely fit the data but learns a graph topology that is both **topologically sparse** and **physiologically interpretable**, bridging the gap between deep learning efficiency and clinical explainability.

## F. Discussion

**Impact and Implications.** Our work demonstrates the potential of integrating State Space Models with domain-specific inductive biases for medical time series analysis. By bridging the gap between the linear computational efficiency of Mamba and the interpretability of adaptive graph learning, MedMamba offers a promising solution for resource-constrained clinical environments.Unlike generic Transformer-based approaches that struggle with quadratic complexity or lack physiological interpretability, our framework successfully uncovers latent functional connectivity (e.g., frontoparietal pathways) while maintaining robustness against non-stationary artifacts like baseline drift. This suggests that lightweight, specialized architectures can outperform general-purpose large models in high-stakes healthcare applications, paving the way for real-time, explainable diagnostic monitoring.

**Limitations and Future Work.** Despite these advancements, our current framework operates primarily in a supervised manner, relying on high-quality annotated data which can be scarce in medical domains. Furthermore, while the learned graph topology provides sample-level explanations, the causal validity of these connections requires further clinical verification. Future work will focus on two directions: (1) exploring self-supervised pre-training strategies to leverage vast amounts of unlabeled physiological data for developing medical foundation models, and (2) extending the architecture to multimodal settings, integrating time series with medical imaging or clinical text to support more comprehensive decision-making.

