# OpenReview forum: "MedMamba: Multi-View State Space Models with Adaptive Graph Learning for Medical Time Series Classification"
_ICML.cc/2026/Conference — ICML 2026 regular_

### Official Review · Reviewer_g4hw · 2026-03-10

**Soundness:** 3
**Presentation:** 3
**Significance:** 3
**Originality:** 3
**Overall Recommendation:** 5
**Confidence:** 4

**Summary:**

This paper studies medical time-series classification, with the goal of handling long sequences, non-stationary drift, and dynamic inter-channel dependencies in a unified and efficient framework. The authors propose MedMamba, which uses a state-space backbone augmented with several domain-specific inductive biases tailored to medical signals. First, a Multi-scale Convolutional Embedding (MCE) is used to capture local waveform morphology. Second, a Tri-branch Differential State Space Encoder (TDSSE) processes raw, difference, and frequency views to improve robustness to baseline drift and non-stationarity. Third, an adaptive Spatial Graph Module (SGM) learns sample-specific cross-channel dependency graphs, with sparsity and acyclicity regularization intended to improve stability and interpretability. The method is evaluated on five real-world EEG/ECG datasets, under both subject-dependent and, more importantly, subject-independent settings.

**Compliance With Llm Reviewing Policy:**

Affirmed.

**Final Justification:**

After considering the paper and the authors’ rebuttal, I have updated my recommendation to Accept. While the performance gains are moderate, I believe the contribution is meaningful and relevant to the community. Overall, the paper meets the bar for acceptance.

**Key Questions For Authors:**

1. Please make the comparison with plain/simplified Mamba baselines more prominent in the paper.
2. Can the authors provide stronger evidence for the interpretability claim of SGM?
3. How well do the robustness experiments correspond to real clinical noise and deployment conditions?

**Limitations:**

yes

**Strengths And Weaknesses:**

Strengths：
1. The paper does not restrict itself to easier sample-level splits, but clearly distinguishes between SD and SI evaluation, and correctly treats the subject-independent setting as the more meaningful test of clinical generalization. This is important in medical time series, where subject identity, sensor placement, and baseline characteristics can easily lead to optimistic estimates if the split protocol is not carefully designed. The paper shows a solid understanding of this issue rather than simply reporting favorable numbers.

2. The paper is not just stacking modules without motivation. MCE is intended to capture local morphology, TDSSE is motivated by non-stationarity and drift robustness, and SGM is designed to model sample-specific cross-channel dependencies. These components map clearly onto the challenges stated in the introduction. In particular, the raw/difference/frequency multi-view design in TDSSE and the fusion of graph-diffusion and channel-wise SSM pathways in SGM are both sensible choices grounded in the characteristics of multi-channel medical signals.

3. The paper evaluates on five datasets with multiple metrics and compares against a reasonably strong set of baselines. Importantly, results are averaged over five random seeds with mean and standard deviation, which is more convincing than reporting only single-run best numbers. The ablations are also meaningfully designed: removing TDSSE hurts most on ADFTD/PTB, fixing the graph or removing sparsity/DAG regularization consistently reduces performance on APAVA and TDBRAIN, and the paper additionally includes drift robustness, missing-channel robustness, and efficiency studies.

Weaknesses
1. The contribution mainly comes from integrating several existing ideas—multi-scale convolutions, differential/frequency multi-view encoding, a Mamba-style backbone, and dynamic graph learning—into a coherent framework for medical time series. This is valuable engineering and seems effective in practice, but the paper does not present a particularly sharp new learning principle or theoretical insight that would clearly generalize beyond this application domain. In particular, parts of MCE and TDSSE feel more like well-motivated domain adaptations than fundamentally new modeling ideas.

2. The SGM includes sparsity and DAG regularization, and the paper repeatedly links this to interpretability. However, the manuscript does not provide sufficiently strong interpretability evidence. For instance, I did not see a systematic analysis showing whether the learned graphs align with known physiological structure, functional connectivity patterns, or clinical priors. Nor is there a compelling qualitative case study demonstrating that the DAG constraint yields meaningful explanatory benefit beyond acting as a regularizer. At present, interpretability appears more like a design intention than a fully supported conclusion.

3. Table 2 shows that MedMamba is often best, but not on every metric and every dataset. For example, on PTB there are baselines with higher AUROC/AUPRC; on APAVA, KEMed performs better on AUROC/AUPRC; and on PTB-XL, Accuracy and AUROC are roughly tied or slightly behind. So the fairest characterization is that the method is overall the most robust and consistently competitive, rather than universally dominating all baselines across the board. The current presentation occasionally feels somewhat stronger than the actual evidence warrants.

---

> ### Author Rebuttal · Authors · 2026-03-31
>
> We sincerely thank Reviewer g4hw for the “Weak Accept” rating, recognizing our rigorous evaluation protocol (SI vs. SD), and for the highly constructive feedback. Your insights have significantly improved the clarity and rigor of our paper. Below is our point-by-point response.
>
> **W1: Novelty and Generality**
>
> Although our architecture is heavily motivated by medical data, our mechanism addresses fundamental mathematical bottlenecks in standard SSMs and GNNs.
> Additionally, MedMamba is specifically designed to address three key failure modes in standard sequence models.
> These are not merely domain-specific adaptations, but **structural innovations that generalize to any non-stationary, multivariate continuous time series** (e.g., industrial sensor networks, traffic):
> * **TDSSE:** Standard SSMs fail under severe distribution shifts. As detailed in Appendix C.2, the differential branch theoretically acts as a high-pass filter to suppress non-stationary trends, providing a stable input distribution for the SSM. Meanwhile, the frequency branch breaks the recurrent memory bottleneck by providing a “shortcut” for global spectral context.
> * **SGM:** Learning dynamic, sample-specific DAGs without prior knowledge is a ubiquitous challenge.
> Our continuous optimization approach using the trace exponential condition $(tr(e^A)-C=0)$, combined with asymmetric projections, enables the data-driven discovery of causal-like routing structures in linear time.
> * **End-to-End Joint Optimization**: MCE, TDSSE, and SGM are not static preprocessing steps or interchangeable add-ons; they are dynamically generated and jointly optimized within the network. Ablations (Section 5.3, Tables 3 & 6) prove these components are essential, not decorative.
>
> We will expand the introduction to clarify this re-framed contribution and explicitly highlight these broader learning principles.
>
> **W2, Q2: Evidence for SGM Interpretability**
>
> We fully agree that claims of interpretability require concrete physiological alignment.
> Due to space constraints in the main text, we relegated the systematic clinical analysis to **Appendix E.9 (Figure 13)**.
> * **Clinical Alignment:** We mapped the adjacency matrix learned on the APAVA dataset to the standard 10-20 EEG system, successfully revealing biologically plausible functional clusters (e.g., O1/O2 are tightly coupled in the visual cortex).
> * **Long-Range Pathways:** SGM identifies specific projections from the parietal/temporal lobes to the frontal lobe, which are recognized clinical biomarkers of cognitive decline in Alzheimer’s disease.
> * **Necessity of DAGs:** Appendix B.2 explains the role of DAG priors in suppressing cyclic amplification and redundant feedback. Furthermore, as shown in Figure 13(b), without our DAG ($\lambda_{DAG}$) and sparsity ($\lambda_{SP}$) priors, the graph collapses into an uninterpretable, oversmoothed “checkerboard.”
>
> **W3: Performance Claims**
>
> We appreciate your meticulous inspection of Table 2. You are absolutely correct that baselines like KEMed, iTransformer, and MedGNN achieve higher AUROC or Accuracy on specific datasets (e.g., PTB, APAVA, or PTB-XL). In the revised version, we will explicitly adopt the precise and fair description: MedMamba is the **overall the most robust and consistently competitive** method, completely removing any claims of “dominance.”
>
> **Q1: Mamba Baselines**
>
> We agree that the comparison with plaom Mamba models (Affirm, S-D-Mamba) is crucial. We will move **Table 8 (Appendix E.8)** to **Section 5.2 (Main Results)** to highlight how our domain-specific induction bias outperforms general Mamba adaptation methods.
>
> **Q3: Robustness Experiments**
>
> Our robustness experiments (Section 5.4) rely on mathematically injected perturbations, but they directly simulate highly prevalent clinical conditions:
> * **Baseline Drift (Fig. 7):** Approximates the low-frequency non-stationarity often caused by patient movement or sensor impedance changes.
> * **Missing Channels (Fig. 9):** Mimics imperfect sensor availability, such as sudden hardware failure or loose leads.
>
> Beyond these synthetic perturbations, our strict **Subject-Independent (SI) protocol** serves as our strongest test for real-world deployment, as it simulates severe distribution shifts (covariate shifts) between unseen patients.
> However, we acknowledge that mathematically synthesized noise cannot capture all chaotic clinical artifacts. We will add this limitations to explicitly map our experiments to these clinical phenomena while acknowledging these practical boundaries.
>
> ---
> **Conclusion**
>
> We hope these clarifications, particularly the inclusion of interpretability evidence and the Mamba baseline in the main text, along with the refinement of our claims, will fully address your core concerns. We are committed to incorporating these improvements into the final manuscript. *We sincerely hope you will improve your score if our response meets your satisfaction.* Thanks for your valuable feedback.

---

> > ### Author Rebuttal · Reviewer_g4hw · 2026-04-05
> >
> > Fully resolved. Thank you for answering.

---

> > > ### Author Response · Authors · 2026-04-07
> > >
> > > Thank you for the clarification and for noting that your concerns are fully resolved!

---

### Official Review · Reviewer_MkDu · 2026-03-11

**Soundness:** 3
**Presentation:** 3
**Significance:** 3
**Originality:** 3
**Overall Recommendation:** 4
**Confidence:** 3

**Summary:**

This paper presents an end-to-end network architecture for medical time series classification (e.g., ECG and EEG) based on mamba. The authors claim that existing methods struggle to jointly model local and global dynamics and handle nonstationarities. To address it, the authors propose a mamba model with a different design: multi-scale convolutional embeddings to capture discriminative local morphology, and a multiview encoder to model raw, temporal-difference, and frequency-domain views. Additionally, they design a spatial graph Mamba module to learns  dependency between sparsity and acyclicity. Experiments on multiple real-world datasets with ablations to show its effectiveness.

**Compliance With Llm Reviewing Policy:**

Affirmed.

**Final Justification:**

The authors' response adequately addressed my concerns, especially on originality. So I adjusted the originality score accordingly.

**Key Questions For Authors:**

See my weakness. Overall, this paper is solid, and I like it, and I expect the authors to address my concerns.

**Limitations:**

Yes

**Strengths And Weaknesses:**

### Strengths
- This paper is overall very clear and well-written. The figures look very beautiful and clear, and the equations are also in a good flow.
- The authors clearly address each issue in different parts of their method. For example, TDSSE with theoretical justification for baseline drift, and SGM for cross-channel dependencies.
- The authors have correctly set up the subject-independent evaluation.
- The authors provide very comprehensive ablation studies, different losses, different modules, and efficiency analysis.
- I also pretty like the qualitative analysis for visualization in E.9 for EEG.


### Weaknesses
- While the combination of modules is practically effective for the medical domain, the architecture feels significantly over-engineered to me. I feel the paper reads more like an application-driven pipeline that explicitly includes existing signal processing modules.
- No discussion on overall computational complexity. How complexity for the spatial module?
- Minor: Also, I noted some recent TSC methods in the top venues related to this topic (drift, interpretability, etc), but not discussed or compared.
For example:

[1] Inherently Interpretable Time Series Classification via Multiple Instance Learning, ICLR'24

[2] FIC-TSC: Learning Time Series Classification with Fisher Information Constraint, ICML'25

[3] Abstracted Shapes as Tokens - A Generalizable and Interpretable Model for Time-series Classification, NeurIPS'2024

---

> ### Author Rebuttal · Authors · 2026-03-30
>
> We sincerely thank Reviewer MkDu for the encouraging review, the “Weak Accept” rating, and for recognizing the clarity, comprehensive ablations, and effectiveness of our method. We highly value your constructive feedback. Below is our point-by-point response.
>
> **W1: Over-Engineering**
>
> We understand this concern: explicitly introducing signal processing concepts may make the architecture appear pipeline-like. However, we do not view MedMamba as a hastily cobbled-together signal processing system. Rather, it is **intentionally designed** to address **three key failure modes** in multichannel medical time series: local multi-scale morphology, nonstationarity / baseline drift, and sample-specific cross-channel dependencies. Standard sequence models often struggle on medical signals because they typically assume stationarity and model channels independently. Accordingly, MCE, TDSSE, and SGM are designed as mathematically grounded inductive biases built around the SSM backbone, rather than interchangeable add-on modules or static preprocessing steps.
>
> More importantly, all components of MedMamba are **jointly optimized end-to-end**, rather than forming a traditional pipeline. The raw, differential, and frequency views are all constructed inside the network and modeled collaboratively through learnable fusion; SGM also does not rely on a predefined graph, but dynamically learns a sample-conditioned directed cross-channel dependency matrix $A \in \mathbb{R}^{C \times C}$, which is optimized jointly with the temporal classifier. Therefore, these modules should be understood as targeted inductive biases introduced to address the core modeling challenges of medical signals.
>
> Ablation experiments further demonstrate that these components are necessary rather than decorative.
> Under the SI setting, removing TDSSE causes the largest performance drop; for example, accuracy on APAVA / PTB-XL / TDBRAIN decreases from 84.48 to 80.41, from 73.86 to 70.46, and from 92.13 to 89.12, respectively. Removing SGM also leads to clear degradation, while removing MCE yields smaller but consistent drops. In addition, the full tri-branch model consistently outperforms reduced-view variants, and its advantage becomes larger under injected baseline drift (Section 5.3, Table 3, Table 6). In the final manuscript, we will clarify that **our contribution lies in the principled, end-to-end integration of lightweight, medically motivated inductive biases into a unified framework, rather than in claiming independent novelty for every atomic operator**.
>
> **W2: Computational Complexity**
>
> We apologize if this analysis was not sufficiently prominent in main paper. A comprehensive theoretical analysis of both time and space complexity is already provided in Appendix D.
> * **Overall Space/Memory Complexity:** Table 4 shows that MedMamba reduces the standard Transformer memory requirement from $\mathcal{O}(T^2)$ memory requirement to $\mathcal{O}(T \cdot CD + C^2)$.
> We also empirically validate our low peak GPU memory usage in Section 5.4 (Figure 8).
> * **Spatial Module Time Complexity:** As derived in Appendix D.2, the theoretical time complexity of the SGM module is $\mathcal{O}(T \cdot C^2 \cdot D + C^3)$.
> In medical time series, the number of channels $C$ is typically very small (e.g., 12 to 33 in our datasets), while the sequence length $T$ is large.
> Therefore, the SGM module effectively scales linearly with $T$, avoiding the $\mathcal{O}(T^2)$ computational bottleneck of standard attention.
>
> We will add an explicit guidance to Appendix D in the main paper to ensure that this discussion is not overlooked.
>
> **W3: Related Work**
>
> We sincerely thank you for pointing out these relevant advancements from top-tier conferences.
> We agree that they significantly enrich the discussion on interpretability and robustness:
> * **[1] MILLET** and **[3] VQShape** provide excellent frameworks for *temporal* and *morphological* interpretability through multi-instance learning and tokenized abstract shapes. MedMamba complements these by providing *spatial* interpretability through directed acyclic graph dependencies learned via SGM.
> * **[2] FIC-TSC** addresses distribution drift through elegant Fisher information constraints. MedMamba achieves this structural goal in a different way, explicitly mitigating baseline drift via our Tri-Branched Differential Encoder (TDSSE).
>
> We will expand our related-work section to specifically discuss these three excellent papers and highlight how their perspectives are complementary to our architectural approach.
>
> ---
> **Conclusion**
> We hope that our clarification regarding the necessity of end-to-end design, our detailed complexity analysis, and the inclusion of the suggested references fully address your core concerns. We are committed to incorporating these updates into the final manuscript. *We sincerely hope you will improve your score if our response meets your satisfaction.* Thank you for your valuable feedback.

---

> > ### Author Rebuttal · Reviewer_MkDu · 2026-03-31
> >
> > I have adjusted the Originality score to reflect the response.

---

> > > ### Author Response · Authors · 2026-04-04
> > >
> > > Thank you very much for the update and for your thoughtful reconsideration of our work. We sincerely appreciate your careful reading of our rebuttal and your recognition that our responses have adequately addressed your concerns. We are also grateful for your adjustment of the originality score. Your constructive feedback has been very valuable in helping us better clarify the motivation, positioning, and contribution of the paper.

---

### Official Review · Reviewer_7iaf · 2026-03-13

**Soundness:** 2
**Presentation:** 4
**Significance:** 3
**Originality:** 2
**Overall Recommendation:** 4
**Confidence:** 3

**Summary:**

Identify the problem that existing models struggle to jointly model local-global dynamics, handle non-stationarities like baseline drift, and capture latent channel dependencies. To address these problems, three modules are proposed: multi-scale convolution, tri-branch differential state space encoder, and spatial graph mamba module.

**Compliance With Llm Reviewing Policy:**

Affirmed.

**Final Justification:**

The authors have addressed most of my concerns, and I updated my score to weak accept. However, I remain reserved regarding the motivation and originality of the paper. The problem identified by the authors has been addressed individually, but not together, which is where the novelty of this paper comes in: a synergistic integration of these solutions. This can still be valuable, but in my opinion, not as impactful as a method that introduces a fundamentally new mechanism in how we approach a problem.

**Key Questions For Authors:**

Please see Strengths and Weaknesses

**Limitations:**

Please see Strengths and Weaknesses

**Strengths And Weaknesses:**

Strengths

1.	The paper is well-written and easy to follow.

2.	The ablation studies are interesting, showcasing the robustness of the proposed framework.

Weaknesses

1.	The problems identified have been addressed in existing literature, therefore, the novelty of the framework is limited. For example, capturing local-global dynamics and channel dependencies has been explored extensively in existing literature, see [1] [2] for example. Handling non-stationarities using tri-branch differential view has also been explored in [3]. As the authors have noted themselves, reducing the quadratic computational complexity of transformer has also been explored in the form of state space models for EEG.

2.	Recent interest in medical time series has shifted towards foundation models, which demonstrate a new frontier in terms of performance. It feels necessary to compare against foundation models as baselines, examples include CBraMod for EEG and HuBERT-ECG for ECG. If existing foundation models can outperform MedMamba, why is the proposed framework useful? While it is understood that MedMamba does not do pretraining, but if a pretrained checkpoint is already available, why would one pick MedMamba over existing foundation model? At a minimum, the author could potentially compare against randomly initialized foundation model, as opposed to pretrained checkpoints, to showcase the advantage of the proposed architecture over existing architectures.

3.	For different tasks, the authors report using slightly different variation of the model, with varying number of layers/dimensions. This brings into question the generalizability of the architecture to different tasks and datasets. Could the proposed architecture form a backbone for foundation models?

References

[1] EEG Conformer: Convolutional Transformer for EEG Decoding and Visualization

[2] CBraMod: A Criss-Cross Brain Foundation Model for EEG Decoding

[3] Toward Foundation Model for Multivariate Wearable Sensing of Physiological Signals

---

> ### Author Rebuttal · Authors · 2026-03-31
>
> We appreciate Reviewer 7iaf for recognizing our work's merits and for the constructive feedback, especially regarding foundation models (FMs). Our point-by-point responses follow.
>
> **W1: Novelty**
>
> While local-global dynamics and multi-view processing are explored concepts [1,2,3], MedMamba’s innovation lies in its **architectural paradigm** and **domain-specific mechanism integration**. It provides a customized end-to-end solution to address three major pain points in medical time series (local morphology, baseline drift, and latent channel interactions), which differs significantly from the cited studies.
>
> 1. **Paradigm Shift & Topology (vs. [1] EEG Conformer, [2] CBraMod):** Both [1] (combining CNN with Transformer) and [2] (masked pre-training model based on cross-Transformer) rely on Transformer with quadratic complexity $\mathcal{O}(T^2)$ and Self-Attention, which generates dense, symmetric dependency matrices. In contrast, MedMamba introduces a linear-time $\mathcal{O}(T)$ SSM backbone, drastically reducing computational overhead. Furthermore, our SGM module uniquely introduces sparsity and a DAG constraint to explicitly capture directional physiological propagation (e.g., seizure foci), greatly improving clinical interpretability.
>
> 2. **Dynamic Drift Handling (vs. [3] NormWear):** While [3] uses static multi-view processing primarily for tokenization/pre-training, our TDSSE is an architectural innovation deeply integrated within the SSM. TDSSE dynamically fuses a differential view (a mathematical high-pass filter suppressing baseline drift) and a frequency view, utilizing a learnable Gated Fusion mechanism based on local signal states.
>
> We will revise the manuscript to clearly distinguish **problem motivation** from **architectural novelty**, avoid overstating firstness, and highlight these core differences.
>
> **W2: Comparison with FMs**
>
> We agree with this excellent suggestion. We conducted additional experiments using **official configurations** (including **training from scratch (S)** and **pre-trained (P) checkpoints**) for **CBraMod** (EEG) and **HuBERT** (ECG).
>
> 1. **EEG:**
>
> Model|ADFTD (Acc/F1)|APAVA (Acc/F1)
> -|-|-
> CBraMod (P)|48.63/51.70 |72.12/80.56
> CBraMod (S)|33.33/27.19|74.63/80.80
> MedMamba|**57.56/55.98**|**84.48/83.03**
>
> *Analysis:* ADFTD exhibits severe non-stationarity, which MedMamba naturally handles via TDSSE, whereas CBraMod lacks explicit drift mitigation.
> APAVA is a smaller dataset where fine-tuning a massive FM proves unstable. MedMamba’s inductive biases learn effectively without massive pretraining data.
>
> 2. **ECG:**
>
> Model|PTB (Acc/F1)|PTB-XL (Acc/F1)
> -|-|-
> HuBERT-Small (P)|75.42/74.67|61.17/60.24
> HuBERT-Small (S)|70.74/65.99|53.81/52.50
> HuBERT-Base (P)|77.28/76.24|61.17/60.25
> HuBERT-Base (S)|73.11/69.63|53.79/52.93
> HuBERT-Large (P)|78.40/77.43|67.88/**65.85**
> HuBERT-Large (S)|74.72/74.66|58.37/58.35
> MedMamba|**89.36/87.36**|**73.86**/63.20
>
> *Analysis:* On PTB, MedMamba substantially outperforms all variants of HuBERT. On PTB-XL, although the scaled HuBERT-Large-Pretrained achieves a slightly higher F1 score, MedMamba is more efficient.
>
> 3. **Why choose MedMamba?**
>
> * **Medical Inductive Biases:** FMs treat signals as generic patches. MedMamba explicitly reveals latent, directed physiological topologies via our SGM module, crucial for multi-lead ECG and multi-channel EEG diagnostics.
> * **Efficiency:** FMs require massive compute; MedMamba is highly feasible to deploy in resource-constrained clinical environments.
>
> *(Note: We will add these into final version, and are willing to provide experiment logs in a rule-compliant manner if needed.)*
>
> **W3: Generalizability and FM Potential**
>
> 1. **Generalizability and Capacity Scaling:**
> Our core architecture (MCE+TDSSE+SGM) is identical across all tasks. Varying layers ($L$) and dimensions ($D$) simply adjusts model capacity for dataset size/complexity (akin to scaling ResNet-18 to ResNet-50), following standard practice [MedFormer (NeurIPS 24), MedGNN (WWW 25)]. This design is validated on 5 diverse EEG/ECG datasets (SD/SI protocols), with ablations confirming consistent module contributions.
>
> 2. **Potential as FM Backbone:**
> Our view is that **possible in principle**, but this paper does not make that claim empirically.
> The architecture has several properties that make it a promising backbone candidate: linear complexity in sequence length, explicit handling of nonstationarity, preservation of local morphology, and sample-adaptive cross-channel modeling. Our current study focuses on supervised learning; however, extending MedMamba to a true foundation model through large-scale self-supervised pretraining is a promising future direction.
>
> ---
> **Conclusion**
>
> We earnestly hope our responses can address your core concerns. We are committed to fully integrating these benchmark results and relevant discussions into the final manuscript. *We sincerely request that you consider raising your score.* Thanks for your valuable feedback.

---

> > ### Author Rebuttal · Reviewer_7iaf · 2026-04-03
> >
> > Thank you to the authors for the detailed responses.
> >
> > W1. As stated in the original review, the problem identified by the authors (local-global dynamics and multi-view processing) has been addressed in the literature for medical time series. Therefore, the motivation for why the authors chose to focus on these problems is unclear. Furthermore, while existing works that tackle these problems used CNN-transformer architecture, tackling these problems again using Mamba is not so significant and novel.
> >
> > W2. I appreciate the comparisons. While I understand the use of different metrics and potentially different dataset splits, why does HuBERT-ECG-Small report 90 AUROC for PTB-XL, but only 60 ACC/F1 in your table? It appears to be quite a large gap.
> >
> > W3. Thank you for the clarification.

---

> > > ### Author Response · Authors · 2026-04-04
> > >
> > > We sincerely thank Reviewer 7iaf for the continued engagement.
> > >
> > > **W1: Motivation, Novelty, and the "Mamba Swap" Concern.**
> > >
> > > **We respectfully but firmly disagree that once a clinical problem has been "addressed" in prior work, the motivation for better solutions is unclear; nor is our work a trivial "Mamba swap."** Our motivation and contributions are:
> > >
> > > **1. "Addressed" does not mean "Solved" (Motivation):**
> > > Previous CNN-Transformer architectures have attempted to model local-global dynamics and multi-view processing, they do so at a prohibitive $\mathcal{O}(T^2)$ cost and typically rely on dense, symmetric attention. In real clinical settings requiring continuous, long-term monitoring (e.g., 24-hour Holter or prolonged EEG), Transformers face catastrophic memory bottlenecks. Thus, the problem of robust, **long-term multi-view modeling under strict clinical computational constraints remains unresolved**. This is our core motivation.
> > >
> > > **2. Beyond a Trivial "Mamba Swap" (Novelty):**
> > > **MedMamba is not a plug-and-play use of vanilla Mamba**. Standard SSMs struggle with the severe non-stationarities and complex channel topologies of medical time series. We did not simply revisit these problems with Mamba; instead, **we introduced novel, domain-specific mathematical mechanisms into the SSM formulation**:
> > > * **TDSSE** is not a standard SSM; it mathematically embeds a differential view (acting as a physical high-pass filter) and a frequency modulation branch directly into the state-space encoding process via a dynamic Gated Fusion mechanism. This physically grounded drift-suppression cannot be achieved by standard Attention or vanilla Mamba.
> > > * **SGM** departs from the symmetric attention maps of Transformers by enforcing a **DAG** constraint. This allows the model to learn the directional propagation of physiological signals (e.g., tracing a seizure focus), providing a level of clinical interpretability that Transformers inherently lack.
> > >
> > > **3. Alignment with Top-Tier Community Standards:**
> > > The reviewer implies that pursuing specialized architectures for these problems may be less significant than foundation models. However, **the broader community clearly recognizes the necessity of advancing domain-customized, non-foundation models for medical time series**. Recent works like **MedFormer (NeurIPS '24), MedGNN (WWW '25), TarDiff (KDD '25), and DynaGraph (npj Digital Med '26)**, also tackle multi-scale and topological problems using specialized, non-foundation architectures. They are valued precisely because they address the unique inductive biases and strict deployment constraints of medical data. **MedMamba advances this line of research** by offering the first linear-time $\mathcal{O}(T)$ SSM specifically designed to address the unresolved bottlenecks of earlier Transformer-based attempts.
> > >
> > > **W2: Numerical gap between AUROC and ACC/F1**
> > >
> > > The large numerical gap does exist, but this is not due to biased reproduction; **it reflects the mathematical behavior of different metrics on extremely imbalanced, multi-label medical datasets.** PTB-XL is a multi-label benchmark with 71 potential labels and 6 evaluation subsets; due to label imbalance and multi-label complexity, AUROC is the primary metric.
> > >
> > > To ensure fairness, **we fully reproduced HuBERT-ECG on PTB-XL using the official settings and uploaded complete test log figures via an anonymous link (https://log-png.pages.dev/anonymous_log_viewer)**:
> > > HuBERT|AUROC|ACC/F1
> > > -|-|-
> > > S|88.86|61.17/60.24
> > > B|88.50|61.17/60.25
> > > L|86.30|67.88/65.85
> > >
> > > **Why is there such a large gap in the same test?**
> > >
> > > 1. **Our reproduction closely aligns with the original paper**: As shown above, our reproduced AUROC for HuBERT-Small is 88.86, close to the 90.00 reported (Supplementary Table 8). This difference may be due to the experimental environment and random seed. **This shows that our setup strictly followed the official implementation and did not deliberately suppress the baseline.**
> > > 2. **Mathematical differences between metrics (AUROC vs. ACC/F1):** PTB-XL is a multi-label classification dataset with extreme class imbalance.
> > >    * **AUROC is threshold-independent:** It measures the model's "ranking ability" (placing positive samples above negative ones). In highly imbalanced datasets, the massive number of True Negatives mathematically inflates the AUROC, creating an illusion of near-perfect performance.
> > >    * **ACC and F1 are threshold-dependent:** They require a hard classification threshold (e.g., 0.5) to make a definitive decision. Even if a model ranks well (yielding a high AUROC), poor probability calibration will prevent predictions from correctly crossing the threshold. Furthermore, Macro-F1 heavily penalizes misclassifications on minority classes, leading to a steep drop in its value compared to AUROC.
> > >
> > > We hope these clarifications address the reviewer’s concerns. *If the reviewer finds them satisfactory, we would greatly appreciate reconsideration of the score.*

---

### Decision · Program_Chairs · 2026-04-30

**Decision:**

Accept (regular)

**Comment:**

After the author/reviewer discussion, the reviewers unanimously favored acceptance, and from carefully reading over the discussion, I am also in agreement in recommending the paper for acceptance.